# Reduced effective radiative forcing from cloud-aerosol interactions (ERF$_{aci}$) with improved treatment of early aerosol growth in an Earth System Model

Sara Marie Blichner[1], Moa Kristina Sporre[2], and Terje Koren Berntsen[1]

[1]Department of Geosciences and Centre for Biogeochemistry in the Anthropocene, University of Oslo, Oslo, Norway
[2]Department of Physics, Lund University, Lund, Sweden

**Correspondence:** Sara Marie Blichner (s.m.blichner@geo.uio.no)

**Abstract.** Historically, aerosols of anthropogenic origin have offset some of the warming from increased atmospheric greenhouse gas concentrations. The strength of this negative aerosol forcing is, however, highly uncertain – especially the part originating from cloud-aerosol interactions. An important part of this uncertainty originates from our lack of knowledge about the pre-industrial aerosols and how many of these would have acted as cloud condensation nuclei (CCN). In order to simulate CCN concentrations in models, we must adequately model secondary aerosols, including new particle formation (NPF) and early growth, which contributes a large part of atmospheric CCN. In this study, we investigate the effective radiative forcing (ERF) from cloud–aerosol interactions (ERF$_{aci}$) with an improved treatment of early particle growth, presented in Blichner et al. (2021). We compare the improved scheme to the default scheme, OsloAero, both part of the atmospheric component of the Norwegian Earth System Model v2 (NorESM2). The improved scheme, OsloAeroSec, includes a sectional scheme that treats the growth of the particles from 5–39.6 nm in diameter which thereafter inputs the particles to the smallest mode in the pre-existing, modal aerosol scheme. The default scheme parameterizes the growth of particles from nucleation and up to the smallest mode, a process that can take several hours. The explicit treatment of the early growth in OsloAeroSec on the other hand, captures the changes in atmospheric condition during this growth time both in terms of air mass mixing, transport and condensation and coagulation.

We find that the ERF$_{aci}$ with the sectional scheme is $-1.16\,\mathrm{Wm}^{-2}$, which is $0.13\,\mathrm{Wm}^{-2}$ weaker compared to the default scheme. This reduction originates from OsloAeroSec producing more particles than the default scheme in pristine, low-aerosol-concentration areas and less NPF particles in high-aerosol areas. We find, perhaps surprisingly, that NPF inhibits cloud droplet activation in polluted/high-aerosol-concentration regions because the NPF particles increase the condensation sink and reduces the growth of the larger particles which may otherwise activate. This means that in these high-aerosol regions, the model with lowest NPF – OsloAeroSec – will have highest cloud droplet activation and thus more reflective clouds. In pristine/low aerosol regions however, NPF enhances cloud droplet activation, because the NPF particles themselves tend to activate.

Lastly, we find that sulphate emissions in the present day simulations increase the hygroscopicity of the secondary aerosols compared to the pre-industrial simulations. This makes NPF particles more relevant for cloud droplet activation in the present day than the pre-industrial atmosphere, because the increased hygroscopicity means they can activate at smaller sizes.

 # 1  Introduction

Since pre-industrial times, humans have significantly shaped our climate through emitting greenhouse gases to the atmosphere. However, the warming induced from these emissions has been masked by the cooling effects of anthropogenic emissions of aerosols and their precursors (Myhre et al., 2013). This cooling is highly uncertain and dominates the spread in estimates of radiative forcing and observationally based estimates of climate sensitivity (Myhre et al., 2013).

The present-day atmospheric aerosols state is challenging to fully characterize due to its fast-changing nature, making point observations hard to generalize. The pre-industrial atmosphere, however, is even more challenging since we cannot rely on direct observations, and thus is only accessible through putting our best knowledge of aerosol processes and sources into models. The pre-industrial atmospheric state is furthermore, very important for estimating the cooling by aerosol cloud inter-actions (Carslaw et al., 2013) because the cloud albedo is more sensitive to perturbations in a "cleaner" atmosphere (Carslaw et al., 2013; Twomey, 1991). There are two main reasons for this. Firstly, cloud droplets form around cloud condensation nuclei (CCN) when the air mass is cooled, normally through adiabatic lifting. The number of particles that will act as CCN and form cloud droplets is dependent on the maximum achieved supersaturation during the cloud formation and how many particles can activate at this supersaturation – which is dependent on size and hygroscopicity. If there are many large CCN, then these will activate "early" during the cloud formation and constitute a water vapor sink which limits the maximum supersaturation and therefore the number of CCN which can activate. We will refer to this effect as supersaturation adjustment. Secondly, cloud albedo $A$ increases with change in cloud droplet number concentration (CDNC) roughly as $dA/d\text{CDNC} = A(1-A)/(3\text{CDNC})$, which entails a lower increase in albedo with a higher baseline CDNC (Twomey, 1991; Carslaw et al., 2013). Therefore, an initial state with higher CCN concentration will be less sensitive to CCN perturbations than an initial state with lower CCN concentrations (Twomey, 1959; Bellouin et al., 2020; Carslaw et al., 2013).

One important, but poorly understood, process for adequately simulating the pre-industrial atmosphere is new particle formation (NPF), i.e. the formation and growth of new particles in the atmosphere which can grow to act as CCN. Roughly speaking, the efficiency of NPF – i.e. how 'many' particles are formed per available condensate – in the pre-industrial atmosphere will determine if the secondary aerosol mass is distributed as very few, very large particles or many smaller particles. Especially in a clean atmosphere, this can play a large role for CCN and CDNC concentrations. Over recent years, the understanding of the drivers of NPF has increased significantly due to improved instrumentation and extensive research (Kerminen et al., 2018; Lee et al., 2019). However, adequately capturing NPF in climate models is difficult due to the requirement for computational efficiency combined with the fine scale of the governing processes, in addition to incomplete scientific understanding of the mechanisms involved (Kerminen et al., 2018; Lee et al., 2019).

NPF starts with the formation of a cluster of molecules which must then activate with respect to the condensing atmospheric vapors and grow into larger sizes ($\sim10\,\text{nm}$ in diameter) (Kerminen et al., 2018; Semeniuk and Dastoor, 2018). Due to the Kelvin effect, few gases have low enough volatility to participate in the very first stages of NPF, while as the particles grow, more gases contribute (Semeniuk and Dastoor, 2018). During this growth, the particles are subject to coagulation with larger particles which constitute a loss in number concentration (Kerminen et al., 2018). The coagulation sink is approximately proportional to

$1/d_p^m$, where $d_p$ is the particle diameter and $m$ is a parameter dependent on the background aerosol concentrations (typically 1.6-1.8) (Lehtinen et al., 2007). It is therefore important for successful NPF that the growth rate (GR) is high enough for the particles to quickly grow to larger sizes where the coagulation sink is lower (Lehtinen et al., 2007). Both Lee et al. (2013) and Olenius and Riipinen (2017) show that omitting explicit modelling of this early aerosol growth and rather parameterizing the survival percentage of particles (e.g. Kerminen and Kulmala, 2002; Lehtinen et al., 2007), lead to significant overestimation of particles. This is mainly because these parameterizations assume steady state conditions during the growth, i.e. that growth rate and coagulation sink are constant, and changes in chemistry, mixing or emissions cannot be taken into account. This assumption is usually not appropriate, especially since the growth can take many hours or even days.

The importance of adequately capturing NPF in modelling the pre-industrial atmosphere is illustrated in a study by Gordon et al. (2016), which shows a major reduction (27 %) in estimated forcing from cloud albedo change when including a nucleation pathway from pure biogenic organics. NPF is subject to several constraints which would indicate more efficient NPF in the pre-industrial atmosphere compared to the present day. Firstly, since the pre-existing aerosol concentrations and thus condensation sink will be lower, the gas phase precursor concentrations are higher *per emissions* than in the present day atmosphere. In other words, if an aerosol precursor species were to have the same emissions/production in the pre-industrial and present day atmosphere, the pre-industrial atmosphere would have higher gas phase concentrations because the condensation sink would be lower. Secondly, the coagulation sink of the clusters and newly formed particles is smaller in a cleaner atmosphere (Carslaw et al., 2013; Gordon et al., 2017).

To better capture the early growth of particles from formation to CCN sizes, we have previously implemented a sectional scheme in the aerosol scheme, OsloAero, of the Norwegian earth system model (Blichner et al., 2021). We refer to the aerosol scheme with the sectional scheme as OsloAeroSec. OsloAeroSec includes 5 bins and 2 condensing species ($H_2SO_4$ and low volatile organics) and treats only the growth/loss of particles from formation at 5 nm and up to the pre-existing modal aerosol scheme at 39.6 nm diameter, in which climate (cloud/radiation) interactions are considered. See Fig. 1 for illustration of the scheme. This work was motivated by 1) the smallest mode in the aerosol scheme OsloAero6 is quite large (number median diameter 23.6 nm), meaning that the growth up to 23.6 nm is parameterized. As mentioned above, this has been shown to lead to significant overestimates of the particle formation (Lee et al., 2013; Olenius and Riipinen, 2017). 2) A sectional scheme explicitly grows the particles and does not a priori assume a shape to the size distribution. In this way it is more physically realistic than including e.g. a nucleation mode. Additionally, the sectional scheme allows for differentiating which organic vapors can contribute to the growth from 5 nm and upwards compared to from nucleation and up to 5 nm.

Our results presented in Blichner et al. (2021), show considerable improvement in the representation of CCN size particles ($> 50$ nm) compared to observations, significantly reducing the frequent high bias in the original model. This goes in line with Olenius and Riipinen (2017) and Lee et al. (2013). On the other hand, the sectional scheme shows an increase in particle number concentrations in remote areas like the polar regions and the free troposphere.

Motivated by both the improvement to the aerosol scheme, and the spatial difference in aerosol formation from the original scheme (remote versus polluted), we here investigate the implications of the growth treatment in OsloAeroSec for the pre-

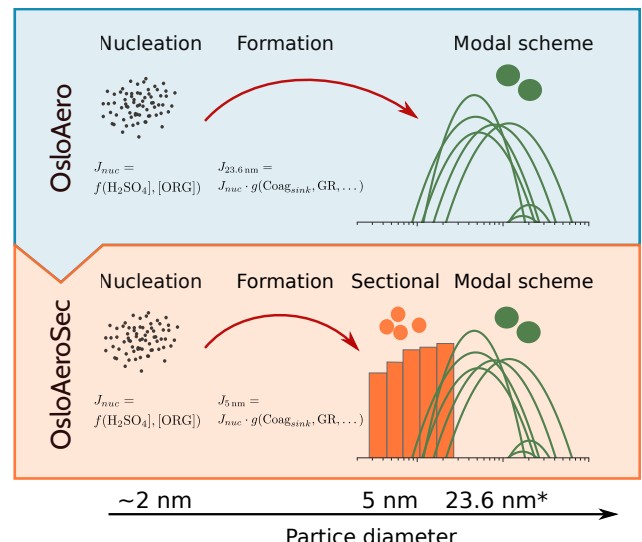

**Figure 1.** Illustration of changes from OsloAero to OsloAeroSec. In both versions, the nucleation rate is calculated at around 2 nm followed by a calculation of the formation rate (the particles surviving) at 5 nm and 23.6 nm in OsloAeroSec and OsloAero respectively, with Lehtinen et al. (2007). In OsloAero, these particles are inserted directly into the modal scheme, while in OsloAeroSec, the particles are inserted into the sectional scheme where they can be affected by growth and coagulation over time and space. Finally, the particles in the sectional scheme are moved from the last bin of the sectional scheme to the modal scheme. *23.6 nm is the number median diameter of the mode the particles from the sectional scheme are moved to, but particles are actually grown to the volume median diameter before they are moved to the modal scheme in order to conserve mass. From Blichner et al. (2021).

industrial and present-day atmosphere respectively and especially for the estimated cooling from aerosol–cloud interactions since pre-industrial times.

The cooling effect is commonly quantified by the radiative forcing (RF) or effective radiative forcing (ERF), which are measures of the change in the net radiation into the atmosphere with adding a climate forcing agent. RF is, by the International Panel of Climate Change's Assessment Report 5 (IPCC AR5) (Boucher et al., 2013) definition, the change in net downwards radiative flux at the tropopause from perturbing the forcing agent, keeping the state variables in the troposphere fixed, but allowing the stratosphere to adjust. However, the ERF is in general considered a better indicator of induced surface temperature

change, because of so called "rapid adjustments" in the atmospheric column which may offset or augment the temperature change from the RF, depending on the forcing agent (Bellouin et al., 2020). In this paper, we therefore use ERF definition as introduced in IPCC AR5, namely the change in top of the atmosphere downwards net flux while allowing adjustments in clouds, temperature, humidity etc. in the atmospheric column, but keeping the sea surface temperature fixed.

## 2 Model description

The Norwegian Earth System Model v2 (NorESM2) (Seland et al., 2020; Bentsen et al., 2013; Kirkevåg et al., 2013; Iversen et al., 2013) is developed with a basis in the Community Earth System Model (CESM) (Danabasoglu et al., 2020; Neale et al., 2012). Firstly, the ocean component, which is not active in these runs since we use fixed sea surface temperature (fSST), is replaced by Bergen Layered Ocean Model (BLOM) (Seland et al., 2020). Secondly, the atmospheric component, CAM6-Nor, differs from the Community Atmosphere Model v6 (CAM6) in CESM in that its aerosol scheme is replaced by OsloAero6 (Kirkevåg et al., 2018) which we describe briefly below.

In this study we investigate the sensitivities of our sectional scheme for early growth which was newly implemented into OsloAero6 by Blichner et al. (2021). Both the original aerosol scheme, referred to as OsloAero, and our version with the sectional scheme implemented, referred to as OsloAeroSec, are described in depth in Blichner et al. (2021). We will therefore only give a brief description of the aerosol scheme here.

All runs are done with CAM6-Nor coupled with the Community Land Model v5 (CLM5) in BGC(biogeochemistry) mode and prognostic crop (Lawrence et al., 2019), prescribed sea ice and sea surface temperatures.

In the following, we start by describing CAM6-Nor in general with the default aerosol scheme, OsloAero, before describing the changes introduced in OsloAeroSec.

### 2.1 CAM6-Nor

As mentioned earlier, CAM6-Nor shares many characteristics with CAM6 (Bogenschutz et al., 2018), while the aerosol scheme exchanged for OsloAero, described below in sec. 2.1.1. The cloud macrophysics are treated with The Cloud Layers Unified by Binormals (CLUBB, Bogenschutz et al., 2013) model. The microphysics for stratiform and shallow convection clouds is the two-moment bulk from Gettelman and Morrison (2015) (MG2), while the deep convection microphysics are treated with a simplified single–moment representation based on Zhang and McFarlane (1995). The cloud activation of aerosols is done with Abdul-Razzak and Ghan (2000). See Bogenschutz et al. (2018) for more details about the clouds.

#### 2.1.1 OsloAero

OsloAero is often referred to as a "production-tagged" aerosol module, meaning that the model to a large extent keeps track of the processes that each tracer has gone through (e.g. coagulation, condensation etc). A key difference to other aerosol modules is that it divides the tracers into "process" tracers and "background" tracers. The idea is that the background tracers decide the number concentration, while the process tracers modify the initial size distribution and chemical composition with a look-up table approach (Bentsen et al., 2013; Kirkevåg et al., 2018, 2013; Iversen et al., 2013; Seland et al., 2020). The background tracers form initial log-normal modes, but after the process tracers are applied, the distribution of the resulting "mixtures" is not necessarily log normal anymore. This distribution is then used for the optical properties and cloud activation.

The chemistry scheme in NorESM uses the preprocessor MOZART (Emmons et al., 2010) to produce a simplified scheme for sulfur and organic species. The oxidant concentrations, hydroxyl radicale (OH), ozone ($O_3$), nitrate radical ($NO_3$) and hydroperoxyl ($HO_2$), are read from file and interpolated from monthly mean. The chemistry scheme treats the oxidation of sulphur dioxide ($SO_2$), dimethyl sulfide (DMS), isoprene and monoterpenes. For a more detailed discussion of the chemistry see Karset et al. (2018), and for a complete overview of reactions and reaction rates, see in particular Table 2 therein.

The aerosol scheme contains three condensing tracers, $H_2SO_4$, and two organic species, namely $SOAG_{LV}$ and $SOAG_{SV}$. The $H_2SO_4$ is produced through oxidation, or emitted directly into the atmosphere. The two organic tracers are produced through oxidation of monoterpene and isoprene, where each reaction has a certain yield of $SOAG_{LV}$ and $SOAG_{SV}$. The reactions of isoprene with $OH, O_3$ and $NO_3$ all yield 5 percent $SOAG_{SV}$, while monoterpene $+ OH$ and monoterpene $+ NO_3$ yield 15 % $SOAG_{SV}$. Finally, monoterpene reacting with monoterpene $+ O_3$ yields 15 % $SOAG_{LV}$, thus being the only reaction yielding $SOAG_{LV}$. The yields used here are similar to those used in other global models (see e.g. Tsigaridis et al., 2014; Sporre et al., 2020; Dentener et al., 2006). All these yields are subject to substantial uncertainty (Shrivastava et al., 2017) – see e.g. Sporre et al. (2020) for an extensive discussion on the sensitivities to these choices.

During condensation these are all treated as non-volatile, but we separate between $SOAG_{LV}$ and $SOAG_{SV}$ because only $SOAG_{LV}$ is considered low-volatile enough to contribute to NPF. In fact only 50 % of the $SOAG_{LV}$ in each time step is assumed to be low enough volatility to contribute to nucleation, and we will refer to this fraction of the $SOAG_{LV}$ as ELVOC.

New particle formation is parameterized by using an intermediate concentration of $H_2SO_4$ and ELVOC in each time step to calculate a nucleation rate followed by a calculation of how many particles survive the growth up to the background mode keeping the particles from NPF (23.6 nm in number median diameter).

The nucleation rate is calculated using Vehkamäki et al. (2002) for binary sulfuric acid-water nucleation and equation 18 from Paasonen et al. (2010) to represent boundary layer nucleation.

This survival of particles from nucleation at $d_{nuc} \approx 2$ nm, the NPF mode is parameterized (number median diameter $d_{mode} = 23.6$ nm) by Lehtinen et al. (2007):

$$J_{d_{mode}} = J_{nuc} \exp\left(-\gamma d_{nuc} \frac{CoagS(d_{nuc})}{GR}\right) \tag{1}$$

where $J_{d_{mode}}$ is the formation rate at $d_{mode}$, $d_{nuc}$ is the diameter of the nucleated particle, $CoagS(d_{nuc})$ is the coagulation sink of the particles [$h^{-1}$], $GR$ is the growth rate [$nmh^{-1}$] of the particle (from $H_2SO_4$ and ELVOC, calculated using eq. 21 from Kerminen and Kulmala (2002)) and $\gamma$ is a function of $d_{form}$ and $d_{nuc}$:

$$\gamma = \frac{1}{m+1}\left[\left(\frac{d_{form}}{d_{nuc}}\right)^{(m+1)} - 1\right], m = -1.6. \tag{2}$$

### 2.1.2 OsloAeroSec

We have implemented a sectional scheme for modelling the growth of particles from nucleation up to the mode which keeps the NPF particles in NorESM (number median diameter 23.6 nm). The scheme is described in detail in Blichner et al. (2021).

The scheme contains five bin sizes set according to a discrete geometric distribution (Jacobson, 2005, sec.13.3) and two condensing vapors: $H_2SO_4$ and $SOAG_{LV}$. The condensation of these species is treated as non-volatile and after condensation, the particles are "grown" (moved) to adjacent bins according to a quasi-stationary structure (Jacobson, 1997, 2005). Coagulation is accounted for both between particles in the sectional scheme and with particles in the modal scheme. When two particles in the sectional scheme coagulate, this contributes to grow the particles, while if they coagulate with particles in the modal scheme, their mass is added to a process tracer in OsloAero (see Blichner et al. (2021) for more details).

The sectional scheme starts at 5 nm and extends to 39.6 nm, where the particles are transferred to the NPF mode in the pre-existing aerosol scheme. The sectional scheme extends to the volume median diameter (39.6 nm) rather than the number median diameter (23.6 nm) in order to preserve both number and mass during the transfer between the schemes.

The boundary layer nucleation parameterization has been updated from Paasonen et al. (2010) to Riccobono et al. (2014), and is now

$$J_{\text{nuc}} = A_3 [H_2SO_4]^2 [\text{ELVOC}] \tag{3}$$

where $A_3 = 3.27 \times 10^{-21}$ cm$^6$ s$^{-1}$.

Finally, in this version of the model, we have also added improvements to the diurnal variation of the oxidant concentrations, described below.

## 2.2 Chemistry: changes to oxidant diurnal variation

The oxidant concentration in CAM6-Nor are read from prescribed 3D monthly mean fields (Seland et al., 2020) with a diurnal cycle superimposed on OH, $HO_2$ and $NO_3$. In the case of OH, this is basically a step function based on before vs after sunrise, which in turn lead to a step function in the $H_2SO_4$ concentration and an unrealistic NPF diurnal cycle. In OsloAeroSec, we therefore implemented a simple sine shape on the daily variation in OH, to improve the realism of NPF.

## 2.3 Model versions

In the result section we compare three different model versions, OsloAero$_{def}$, OsloAero$_{imp}$ and OsloAeroSec. The first, OsloAeroSec, is the default model as used e.g. in the CMIP6 simulations, described in section 2.1.1 above. The second version, OsloAero$_{imp}$, is the default model but with the same changes to the nucleation scheme and the oxidant diurnal variation as are used in OsloAeroSec. The third is with the sectional scheme, OsloAeroSec, as described in section 2.1.2 and by Blichner et al. (2021). This is summarized in Table 1. The motivation for including all these model versions is to be able to distinguish the effect of the sectional scheme from that of the changes in nucleation and oxidants.

**Table 1.** Model version overview.

| Simulation | Nucleation parameterization | Oxidant treatment | Early growth treatment |
|---|---|---|---|
| OsloAeroSec | $A_3[H_2SO_4]^2 \times [ELVOC]$ [*] | Improved diurnal variation | Lehtinen et al. (2007) + sectional scheme |
| OsloAero$_{imp}$ | $A_3[H_2SO_4]^2 \times [ELVOC]$ [*] | Improved diurnal variation | Lehtinen et al. (2007) |
| OsloAero$_{def}$ | $A_1[H_2SO_4] + A_2[ELVOC]$ [†] | Default diurnal variation | Lehtinen et al. (2007) |

$A_1 = 6.1 \times 10^{-7} \text{ s}^{-1}$

$A_2 = 3.9 \times 10^{-8} \text{ s}^{-1}$

$A_3 = 3.27 \times 10^{-21} \text{ cm}^6\text{s}^{-1}$

[*] Riccobono et al. (2014)

[†] Paasonen et al. (2010)

## 3   Simulation setup

All simulations are performed with NorESM2 release 2.0.1 with $1.9°$ (latitude) $\times 2.5°$ (longitude) resolution with 32 height levels from the surface to $\sim 2.2$ hPa in hybrid sigma coordinates. The time step is 0.5 hour. We use a configuration with active atmosphere (CAM6-Nor, Seland et al., 2020) and land component (CLM5-BGC, Lawrence et al., 2019), while sea ice and sea surface temperatures are read from file. We use the fixed SST method combined with nudging to estimate effective radiative forcing (ERF) from aerosol–cloud interaction, $\text{ERF}_{aci}$, and ERF from aerosol-radiation interactions, $\text{ERF}_{ari}$ (Hansen et al., 2005; Forster et al., 2016). This means that we use prescribed SST and sea ice and perturb the anthropogenic aerosol emissions.

We use nudging against model produced meteorology to constrain the natural variability (Kooperman et al., 2012; Zhang et al., 2014; Forster et al., 2016), nudging the horizontal wind components (U,V) and surface pressure with a relaxation time of 6 hours (as described in Karset (2020, sec 4.1)). Only nudging U, V and surface pressure is preferable over nudging more variables (temperature, humidity, energy fluxes, surface drag etc), because it allows for rapid adjustments which should be included in $\text{ERF}_{aci}$. See Karset (2020, ch. 4.1) for discussion.

In addition, we use the method proposed by Karset et al. (2018) to estimate the effective radiative forcing, i.e. we use not only to the anthropogenic aerosol emissions but also the oxidants from the present day atmosphere.

To produce the meteorology, we first ran a 7 years simulation (plus 2 years discarded as spin up), MMET$_{1850}$ with the default model, OsloAero$_{def}$. This was done with standard CMIP6 pre-industrial (here meaning 1850) forcing and emissions.

Two simulations were performed with each model version:

**PI** Pre-industrial (1850) simulation nudged to MMET$_{1850}$

**PD** Simulation with aerosol emissions and oxidant fields from "present day" (2014) nudged to pre-industrial meteorology (MMET$_{1850}$)

**Table 2.** Abbreviations for model configurations and versions.

| | Abbreviation | Description |
|---|---|---|
| Forcing configuration: | | |
| | PI | pre-industrial (1850) run with pre-industrial aerosol emissions and oxidants |
| | PD | pre-industrial (1850) run with anthropogenic emissions and oxidant fields from present day (2014) |
| Model versions: | | |
| | $OsloAero_{def}$ | Run with $OsloAero_{def}$ |
| | $OsloAero_{imp}$ | Run with $OsloAero_{imp}$ |
| | OsloAeroSec | Run with OsloAeroSec |

These are the simulations used to calculate the ERF and which are analyzed in the result section. Emissions of aerosol and precursors for both the present and pre-industrial are from Hoesly et al. (2018); van Marle et al. (2017). Oxidant fields are as described in Seland et al. (2020), from Danabasoglu et al. (2020).

The PI simulations were all initialized from a two-year simulation with $OsloAero_{def}$ model version with pre-industrial conditions and free meteorology (SPINUP_PI). Similarly, the PD simulations, were all initialized from a two-year simulation with $OsloAero_{def}$ model version with free meteorology and pre-industrial conditions but present day aerosol emissions and oxidant fields (SPINUP_PD). MMET_PI, SPINUP_PI and SPINUP_PD were all initialized from a 30-year simulation with PI configuration.

Table 3 summarizes the model simulations and table 2 summarizes the abbreviations for the model versions and configurations.

**Table 3.** Description of runs. See Table 2 for abbreviations.

| | Simulation name | Model version | Forcing conf. | Initialized from | Meteorology | Years |
|---|---|---|---|---|---|---|
| Meteorology: | MMET_PI | $OsloAero_{def}$ | PI | * | Free meteorology | 1–8 |
| Spin-up runs: | SPINUP_PI | $OsloAero_{def}$ | PI | * | Free meteorology | 1–2 |
| | SPINUP_PD | $OsloAero_{def}$ | PD | * | Free meteorology | 1–2 |
| PI runs: | $OsloAero_{def}$_PI | $OsloAero_{def}$ | PI | SPINUP_PI | Nudged MMET_PI | (3)4–8[†] |
| | $OsloAero_{imp}$_PI | $OsloAero_{imp}$ | PI | SPINUP_PI | Nudged MMET_PI | (3)4–8[†] |
| | OsloAeroSec_PI | OsloAeroSec | PI | SPINUP_PI | Nudged MMET_PI | (3)4–8[†] |
| PD runs : | $OsloAero_{def}$_PD | $OsloAero_{def}$ | PD | SPINUP_PD | Nudged MMET_PI | (3)4–8[†] |
| | $OsloAero_{imp}$_PD | $OsloAero_{imp}$ | PD | SPINUP_PD | Nudged MMET_PI | (3)4–8[†] |
| | OsloAeroSec_PD | OsloAeroSec | PD | SPINUP_PD | Nudged MMET_PI | (3)4–8[†] |

[*] 30 year run with PI emissions. [†] Year 3 is discarded as spinup and years 4 to 8 inclusive are used in the analysis.

**Table 4.** Model variable definitions.

| Variable name | Definition |
|---|---|
| $N_a$ | Number of particles excluding those in the sectional scheme |
| $N_{NPF}$ | Number of particles from NPF excluding those in the sectional scheme |
| $N_{d_1-d_2}$ | Number of particles with diameter $d$ such that $d_1 \leq d \leq d_2$ |
| $N_{d_1}$ | Number of particles with diameter $d$ such that $d_1 \leq d$ |

## 4 Terminology

Because we are comparing model versions with and without the sectional scheme, we will only discuss particle number concentrations of particles in the modal OsloAero part of the scheme, that is excluding the ones still in the sectional scheme. This gives us an apples-to-apples comparison with the original model version. We will use $N_a$ to refer to total aerosol concentration, excluding the particles in the sectional scheme, and $N_{NPF}$ for the subset of these particles originating from NPF. Furthermore, we use $N_{d_1-d_2}$ to refer to the particles with a diameter larger than $d_1$ but smaller than $d_2$. These definitions are summarized in Table 4.

We will use the term *NPF efficiency* or *the efficiency of NPF* to describe model to model differences in how many NPF particles are produced with the same emissions (PI or PD). If model version A and B are both run with the same setup (e.g. pre-industrial emissions), and model A produces more NPF particles than model B, we will say that A has higher NPF efficiency than B.

We use the Ghan (2013) method for calculating $ERF_{aci}$ and $ERF_{ari}$, meaning that we output the net radiation at the top of the atmosphere, $F$, and in addition output calls to the radiation scheme with clean (no aerosols), $F_{\text{clean}}$ and clean and clear (no aerosol, no clouds), $F_{\text{clean,clear}}$. Thus, the direct aerosol radiative effect is $DIR_{Ghan} = F - F_{\text{clean}}$ and the cloud radiative effect is $CRE = F_{\text{clean}} - F_{\text{clean,clear}}$. It follows further that $ERF_{ari} = \Delta DIR_{Ghan} = \Delta(F - F_{\text{clean}})$ and $ERF_{aci} = \Delta CRE = \Delta(F_{\text{clean}} - F_{\text{clean,clear}})$, where $\Delta$ signifies the difference between PD and PI.

## 5 Results and discussion

We will start by presenting globally averaged $ERF_{ari}$ and $ERF_{aci}$ in the model versions, and how these relate to PI to PD changes in globally averaged aerosol and cloud properties (section 5.1). Next, in section 5.2, we present a series of hypothesis for the differences in $ERF_{ari}$ and $ERF_{aci}$ between the model versions, which we will use to analyze the results.

In section 5.3, we discuss the PI to PD changes on a regional level, before discussing the PI and PD simulations separately in sections 5.4 and 5.5. We discuss all model versions where this is helpful for understand the results, but we otherwise focus on OsloAeroSec versus $OsloAero_{def}$, because $OsloAero_{def}$ is the version used in CMIP6.

### 5.1 Global averages: Aerosol number and ERF

#### 5.1.1 Aerosol number

In general, the sectional scheme produces more particles than the original scheme in very pristine environments, while producing fewer in areas with high aerosol concentrations (Blichner et al., 2021). This is reflected in the globally averaged profiles of NPF particles, $N_{\text{NPF}}$, for each model version shown in Fig. 2. In the PD simulations, OsloAeroSec mostly has lower $N_{\text{NPF}}$ concentrations than the other model versions, surpassing $OsloAero_{imp}$ only above $\sim 650\,\text{hPa}$. However, in the cleaner PI atmosphere, OsloAeroSec has $N_{\text{NPF}}$ concentrations closer to, or even higher, than the other two schemes. OsloAeroSec has higher $N_{\text{NPF}}$ concentrations above $\sim 850\,\text{hPa}$ and $\sim 700\,\text{hPa}$ compared to $OsloAero_{imp}$ and $OsloAero_{def}$, respectively. Close to the surface, where aerosol concentrations in general are higher, OsloAeroSec has lower $N_{\text{NPF}}$ that the other two models, even in the PI simulation.

As we shall explain more in depth later, these changes in NPF in clean remote versus higher aerosol concentration areas, are important for $ERF_{aci}$ because the NPF particles are more likely to activate in pristine regions, while may even act to suppress activation in the more polluted regions.

Furthermore, note that even though $OsloAero_{imp}$ is the same as OsloAeroSec, excluding the sectional scheme, the profile is qualitatively different: OsloAeroSec has fewer particles close to the ground and much more further up in the PI atmosphere, see section 5.6.

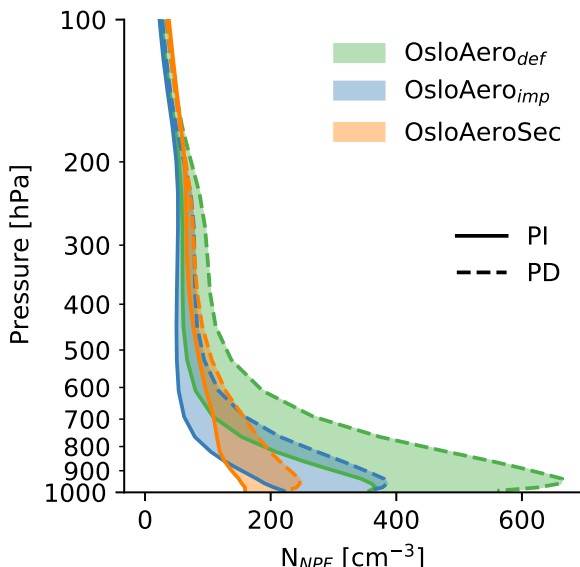

**Figure 2.** Globally averaged concentration of aerosols from NPF. The solid lines show the concentration in the PI simulation, while the dashed lines show the concentration in the PD. The shading signifies the change in each model. Note that the inter-annual variability in the globally averaged $N_{NPF}$ within each simulation is very low (see Fig. S8.

### 5.1.2 ERF

The globally averaged $ERF_{aci}$ is significantly influenced by the introduction of the sectional scheme, as is seen in Fig. 3 showing total, shortwave and longwave components of $ERF_{aci}$, and $ERF_{ari}$. $ERF_{aci}$ in OsloAeroSec is significantly (p< 0.01) lower than both OsloAero$_{def}$ and OsloAero$_{imp}$, using a two-tailed paired Student's t–test on the globally averaged monthly output. The $ERF_{aci}$ is $0.13 \, \mathrm{Wm}^{-2}$ weaker in OsloAeroSec compared to OsloAero$_{def}$. The $ERF_{aci}$ with OsloAero$_{imp}$ and OsloAero$_{def}$ is roughly the same (difference of $0.01 \, \mathrm{Wm}^{-2}$). Also, the total radiative effect from aerosols, $ERF_{aci+ari}$, is lower $\sim 0.1 \, \mathrm{Wm}^{-2}$ in OsloAeroSec compared to both OsloAero$_{def}$ and OsloAero$_{imp}$. One can further see in Fig. 3, that the difference in the $ERF_{aci}$ between the OsloAeroSec and OsloAero$_{def}$ is completely caused by difference in the SW forcing. Moreover, even though OsloAero$_{imp}$ has roughly the same $ERF_{aci}$ as OsloAero$_{def}$ (not significantly different with p< 0.05) it has a significant strengthening (p< 0.01) of the forcing in both the SW and LW component that ends up cancelling each other out in the total forcing. Lastly, the direct effective aerosol forcing, $ERF_{ari}$, is also shown in Fig. 3 and the direct effect is slightly closer to zero with OsloAeroSec than OsloAero$_{def}$ and OsloAero$_{imp}$ ($\sim$-0.03 $\mathrm{Wm}^{-2}$ smaller than OsloAero$_{def}$ and OsloAero$_{imp}$, significant with p< 0.01). It may seem surprising that both OsloAero$_{def}$ and OsloAero$_{imp}$ have positive $ERF_{ari}$. Note that we are using Ghan (2013) to calculate $ERF_{ari}$ and that other methods may give a slightly different result. Smith et al. (2020) show comparisons of different estimates of the $ERF_{ari}$ for CMIP6 models and find similar values to ours

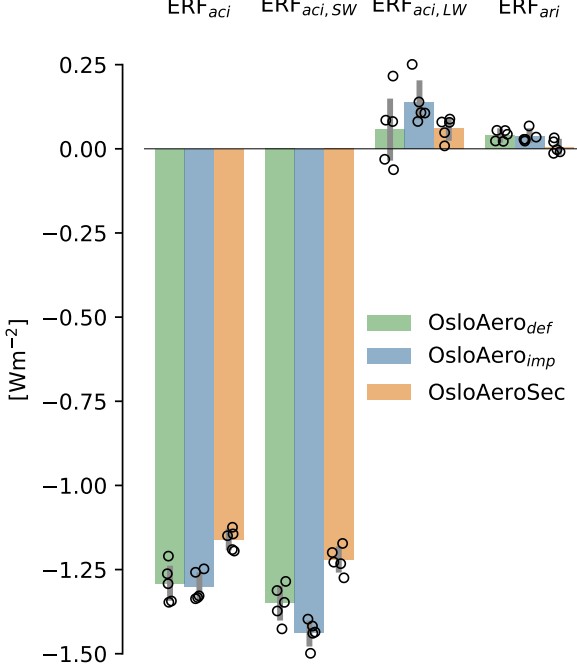

**Figure 3.** Globally averaged effective radiative forcings (ERF) from aerosols. $ERF_{aci}$ is the ERF from aerosol-cloud interaction, $ERF_{aci,SW}$ and $ERF_{aci,LW}$ are the short wave and long wave component of $ERF_{aci}$ and $ERF_{ari}$ is the ERF from aerosol radiation interaction alone. All are computed in accordance with Ghan (2013). The circles are the the averages for each individual year in the 5 year simulations and the gray bar indicates the 95% confidence interval of the mean.

for NorESM with the Ghan (2013) method, while e.g. the approximate partial radiative perturbation (APRP) method while the APRP method gave a negative $ERF_{ari}$ for the same simulations. The difference between OsloAeroSec and the default model likely originates from OsloAeroSec producing fewer particles than $OsloAero_{def}$ in the PD simulation and thus allowing the remaining particles to grow larger and thus scatter radiation more efficiently (Blichner et al., 2021).

As discussed in the introduction, $ERF_{aci}$ depends both on the increase in CCN between PI and PD and on the number of CCN in the PI base state. The less CCN there is in the base state, the larger the impact of a given increase in CCN will be, because the clouds are more susceptible. As OsloAeroSec has much lower particle number concentrations than $OsloAero_{def}$ in the PI, we might expect OsloAeroSec to have a less CCN/CDNC and weaker (less negative) $NCRE_{Ghan}$ in the PI. In this case OsloAeroSec would have clouds that are more susceptible to change from PI to PD, than $OsloAero_{def}$. The opposite is in fact the case, as can be seen in Fig. 4 which relates the column burden of $N_{NPF}$ particle mass (which, due to the technical setup of OsloAero, is proportional to the number) to the net cloud radiative effect ($NCRE_{Ghan}$). While the column burden of $N_{NPF}$ is lower in OsloAeroSec compared to $OsloAero_{def}$, the $NCRE_{Ghan}$ is stronger (more negative). On the other hand, $OsloAero_{imp}$ has the lowest column burden of $N_{NPF}$ and the weakest $NCRE_{Ghan}$, and thus follows the logic that a "cleaner" atmosphere

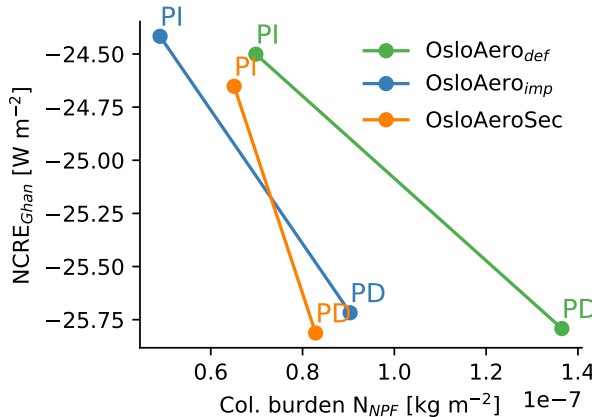

**Figure 4.** Globally averaged aerosol values of $NCRE_{Ghan}$ (y-axis) and column burden of NPF particles (x-axis) for the pre-industrial (PI) and present day (PD) atmosphere. The circles show each annual average and are included to indicate the variability.

gives a less negative (weaker) $NCRE_{Ghan}$. In the PD simulations, OsloAeroSec has the lowest column burden of $N_{NPF}$ of all the models and approximately the same $NCRE_{Ghan}$ as OsloAero$_{def}$, while OsloAero$_{imp}$ has a less negative $NCRE_{Ghan}$

than the other two. Since $ERF_{aci} = NCRE_{Ghan,PD} - NCRE_{Ghan,PI}$, it is clear from Fig. 4, that most of the difference between the schemes originate in different $NCRE_{Ghan}$ in the PI simulations; $-0.15$ and $-0.24\,\mathrm{Wm}^{-2}$ compared to OsloAero$_{def}$ and OsloAero$_{imp}$, respectively. The difference in the PD simulations partially compensate this but is considerably smaller; $-0.02$ and $-0.1\,\mathrm{Wm}^{-2}$ compared to OsloAero$_{def}$ and OsloAero$_{imp}$, respectively. Furthermore and maybe surprisingly, this plot shows that the change in $NCRE_{Ghan}$ per change in column burden $N_{NPF}$ (i.e., the slope of the line in Fig. 4), is much more

negative for OsloAeroSec than for the other two model versions.

### 5.2 Reasons for differences in $ERF_{aci}$

From what we have seen so far, it is first of all clear that changes in the PI $NCRE_{Ghan}$ are dominating the difference in $ERF_{aci}$ between the models, i.e. the spread in modelled $NCRE_{Ghan}$ between the models is larger in PI than in PD. Secondly, we have seen that at least in globally averaged properties, more efficient NPF, meaning more particles with the same emissions, does not

necessarily lead to a stronger negative $NCRE_{Ghan}$. To explain the somewhat unintuitive relationship between particle number and $NCRE_{Ghan}$, we must consider also their geographical distributions with respect to where the NPF particles are likely to activate in clouds and contribute to CDNC. In this section we first outline some important processes and then layout some hypothesis for the difference in $NCRE_{Ghan}$ with OsloAeroSec compared to the other versions. These will serve to ease the rest of the results and discussion.

The cloud droplet activation of particles and resulting CDNC depend on the following factors: 1) The maximum achieved supersaturation ($S_{max}$) together with the hygroscopicity of the particles decide the activation diameter of each mode, 2) $S_{max}$

depends on the updraft velocity, but is also influenced by supersaturation adjustment due to the uptake of water vapor from large(r) particles which activate "early" during lifting, and finally, 3) the absolute number of particles in each mode which are larger than the activation diameter and thus activate.

Furthermore, note that the number of particles from NPF is strongly negatively correlated with the number median diameter of the modes in the size distribution, both the NPF mode and the larger modes. This is because the total available surface area is larger when there are more NPF particles, which means the available condensate is distributed to more numerous, but smaller particles. This leads, as we will show, to NPF inhibiting cloud droplet activation in many regions in the model.

Figure 5 illustrates the effect of changing the NPF efficiency on CDNC in two different environments. For simplicity, let us
assume that we are comparing two models with different NPF efficiency; model A with high NPF efficiency and model B with low NPF efficiency. As noted above, model A will have more numerous, but smaller, particles (A1 and A2 in Fig. 5), while model B will have fewer, but larger particles (B1 and B2 in Fig. 5). Furthermore, we will consider two different environments. Environment 1 has a small activation diameter because, e.g. there are few large particles (no early activation) or the updraft is strong (A1 and B1 in Fig. 5). Environment 2 has a large activation diameter because, e.g. it has high emissions of large primary
particles which activate early and limit the maximum supersaturation (A2 and B2 in Fig. 5). In this simplification we assume that the activation diameter does not change between model A and B. This is not strictly true, but a good assumption because the inter-model changes in $S_{max}$ (Fig. S21) and hygroscopicity (Fig. S26) are small and do not dominate the response in terms of CDNC.

We start with environment 1 where the activation diameter is small (e.g. Antarctica). This is illustrated by the two size
distributions, A1 and B1, on the top in Fig. 5. In this environment model A (high NPF efficiency, A1) will result in higher cloud droplet activation and higher CDNC than model B (low NPF efficiency, B1). This is because a considerable fraction of the small NPF mode particles activate, and thus the decrease in the size of the larger particles does not matter.

Next we consider environment 2 where the activation diameter is large (e.g. a polluted area like China).This is illustrated by the two size distributions, A2 and B2, at the bottom of in Fig. 5. In this environment model A with high NPF efficiency (A2)
will result in lower cloud droplet activation and lower CDNC than model B with a low NPF efficiency (B2). This is because the change in the diameter of the larger particles is the only thing which is matters for activation, since the smaller particles will not activate anyways.

In this simplified thought example, we can say that in environment 1 (small activation diameter), NPF enhances cloud droplet activation while in environment 2 (large activation diameter), NPF inhibits cloud droplet activation.
With all this in mind, we can lay out some plausible hypothesis that might contribute to a weaker $ERF_{aci}$ in OsloAeroSec compared to the other model versions:

1. **Smaller $\Delta_{PD-PI}N_a$:** The difference in $ERF_{aci}$ is due to a smaller change in number concentration between PI and PD in OsloAeroSec than the other model versions

2. **Higher $N_a$ in PI:** OsloAeroSec produces more particles under PI conditions and therefore the clouds are less susceptible
to increased anthropogenic emissions

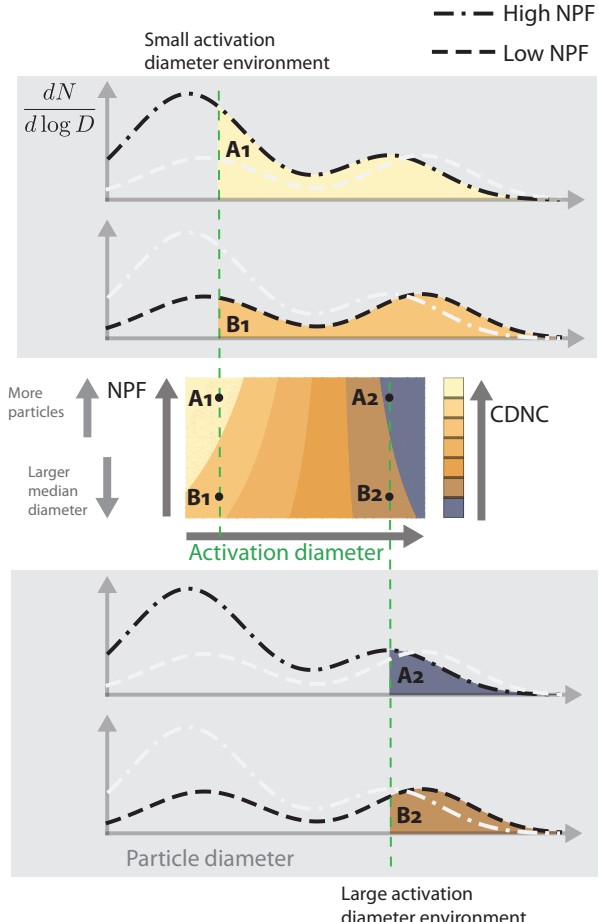

**Figure 5.** Schematic illustrating the influence of NPF on cloud droplet activation and CDNC. The top and bottom panel shows what happens to activation in two different environments (1 and 2) and for two models; one model with high NPF efficiency (A) and one with low NPF efficiency (B). Let us first consider environment 1 (top panels): here the activation diameter is small (either due to strong updrafts, few large particles or high hygroscopicity) and particles all the way down to the mode holding the NPF particles ($\sim$ Aitken mode) activate. In this environment model A will activate more particles than model B and have higher CDNC. Next let us consider environment 2 (bottom panels): here the activation diameter is large (due to weak updrafts, supersaturation adjustment due to larger particles or hygroscopicity) and only the largest particles activate. Here model B will activate more particles than model A because the size of the larger particles is what dominates.

3. **Higher activation in PI:** The number of particles that actually act as CCN and activate is higher with OsloAeroSec than the other model versions in the PI simulations, leading to a higher baseline CDNC. This is due to

    (a) more efficient NPF in remote regions where NPF enhances activation

    (b) less efficient NPF in regions where NPF inhibits activation (only larger particles activate)

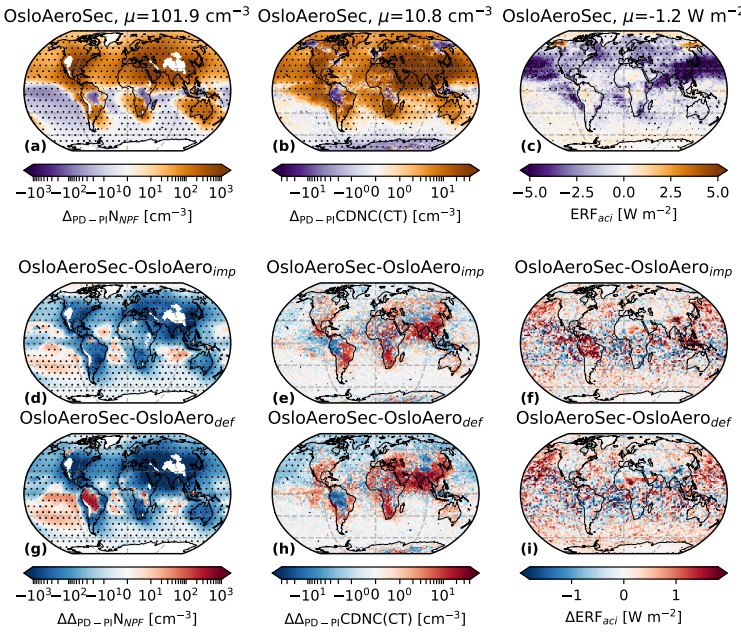

**Figure 6.** Annual average values of near surface $N_{NPF}$ concentrations (left column), cloud top droplet number concentration (CDNC(CT), middle column) and $NCRE_{Ghan}$ ( right column). The top panel shows the PD - PI for OsloAeroSec while the second and third rows show the change in this value (PD-PI) from $OsloAero_{imp}$ (second row) and $OsloAero_{def}$ (third row) to OsloAeroSec. The $N_{NPF}$ values are are averaged up to 850 hPa and weighted by pressure thickness of each grid cell. Dots are included in the plots to indicate where the difference between the two models is significant with a two-tailed paired Student's t–test with 95 % confidence interval.

4. **Lower activation in PD:** The number of particles that actually act as CCN and activate is lower with OsloAeroSec than the other model versions with PD emissions, leading to a weaker $ERF_{aci}$. This is

     (a) due to lower NPF efficiency regions where NPF enhances activation

Hypothesis 2 has partly already been disproven because in terms of global averages, $OsloAero_{def}$ has higher particle number concentrations than OsloAeroSec all the way up to approximately 700 hPa (with most of the liquid clouds being below this
level).

### 5.3   Pre-industrial to present day changes

We start by considering hypothesis 1, and how the PI to PD change looks on a regional level in OsloAeroSec versus $OsloAero_{def}$.

This is shown in Fig. 6 where the first row is the change between PD and PI ($\Delta_{\text{PD-PI}}$) for OsloAeroSec and the two
subsequent rows are the difference to this first quantity, $\Delta_{\text{PD-PI}}$, between the model versions ($\Delta_{\text{PD-PI}}$(OsloAeroSec) minus
$\Delta_{\text{PD-PI}}$(OsloAero$_{imp}$) and $\Delta_{\text{PD-PI}}$(OsloAeroSec) minus $\Delta_{\text{PD-PI}}$(OsloAero$_{def}$), denoted $\Delta\Delta_{\text{PD-PI}}$). The first column, showing
the near surface averaged $N_{\text{NPF}}$, shows that, as expected, most of the PI to PD change happens in the northern hemisphere.
This is consistent with the major anthropogenic emission sources being located here. Over ocean regions in the Southern
Hemisphere, there is even a small decrease in NPF particles many places. Comparing to OsloAero$_{def}$ (row 3) we see that
OsloAeroSec has a smaller increase in $N_{\text{NPF}}$ from PI to PD, except in the South Pacific and over the Amazon. Especially high
pollution areas over land stand out as strongly negative. Note that the first column in Fig. S15 shows the same but for zonal
averages, and underlines that $\Delta_{\text{PD-PI}}N_{\text{NPF}}$ is higher in OsloAero$_{def}$ than OsloAeroSec all through the atmospheric column.

The second column shows the change in cloud droplet number concentration at cloud top (CDNC(CT)). Again the first
row shows $\Delta_{\text{PD-PI}}$CDNC(CT), which, as expected, shows an increase – in particular in the northern hemisphere. Comparing
OsloAeroSec to OsloAero$_{def}$ (row 3) however, the first thing that stands out is that, somewhat surprisingly, $\Delta\Delta_{\text{PD-PI}}$CDNC(CT)
is positive over polluted regions, meaning that the PI to PD increase in CDNC(CT) is stronger with OsloAeroSec than with
OsloAero$_{def}$, in spite of $N_{\text{NPF}}$ increasing less with OsloAeroSec. In other words, in these regions we are in the bottom panel
of Fig. 5, where more particles are added with OsloAero$_{def}$ than OsloAeroSec, but fewer of these extra particles are activating
into cloud droplets. Meanwhile, in more remote regions, like the North Pacific and the Arctic, we are in the top panel of Fig. 5
and CDNC(CT) increases less with OsloAeroSec than OsloAero$_{def}$, following the more expected logic that a smaller increase
in particle number lead to a smaller increase in cloud droplets from PI to PD.

Finally, the last column shows the ERF$_{aci}$. Here we see (first row, c), that the ERF$_{aci}$ is strongly negative over the North
Pacific as well as over China and India. The difference in ERF$_{aci}$ between the models shows that the remote Pacific dominates
in making ERF$_{aci}$ more strongly negative in OsloAero$_{def}$ than in OsloAeroSec. Even though the increase in CDNC(CT) from
PI to PD is stronger in polluted regions with OsloAeroSec, these regions seem to have reached saturation with respect to
changing albedo and the ERF$_{aci}$ changes little between the model versions.

To summarize with regard to hypothesis 1: the change in particle number between PI and PD is indeed smaller with
OsloAeroSec than the other model versions, but this can only explain the change in CDNC in remote regions (North Pa-
cific, Siberia etc). Furthermore, as mentioned earlier, we need to consider the influence of the baseline aerosol state in PI, and
not just the change between PI and PD.

## 5.4 The pre-industrial atmosphere: model to model differences

To consider hypothesis 3, "Higher activation in PI", we now consider differences between OsloAeroSec and the default model
versions in the PI separately from PD (covered in the next section).

Figure 7 shows the near surface concentration of $N_{\text{NPF}}$ in the PI simulation (left column) for OsloAeroSec (a) and the
relative difference in this value between the model versions (b and c). We see that compared to OsloAero$_{def}$, $N_{\text{NPF}}$ is lower in
OsloAeroSec almost everywhere in PI. However, as is seen in Fig. 8c, showing the zonally averaged difference, this decrease

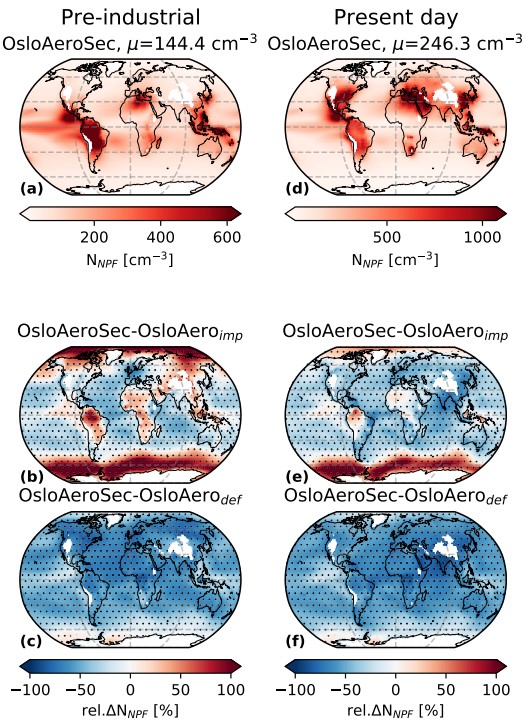

**Figure 7.** Top row: Annual average near surface $N_{NPF}$ concentrations for OsloAeroSec for PI (left) and PD (right). Rows 2–3: the relative difference of OsloAeroSec to OsloAero$_{imp}$ (row 2) and OsloAero$_{def}$ (row 3) for PI (left) and PD (right), respectively. All values are averaged up to $850\,\mathrm{hPa}$ and weighted by pressure thickness of each grid cell. Dots are included in the plots to indicate where the difference between the two models is significant with a two-tailed paired Student's t–test with 95 % confidence interval.

with OsloAeroSec is mostly confined to the near-surface areas. The decrease in $N_{NPF}$ with OsloAeroSec near the surface switches to an increase higher up in the atmosphere.

### 5.4.1 Cloud properties

OsloAeroSec has a higher cloud droplet number concentration at cloud top (CDNC(CT)) than OsloAero$_{def}$ in most of the PI atmosphere, as can be seen in Fig. 9a. This is despite that OsloAeroSec has lower $N_{NPF}$ concentrations in most near-surface areas compared to OsloAero$_{def}$. We must therefore investigate what happens to the size distribution, rather than just the absolute number. Figure 9c, e and g, shows the OsloAeroSec to OsloAero$_{def}$ difference in number concentrations of $N_{100}$, $N_{150}$ and $N_{200}$. The $N_{100}$ concentration (c), is lower in OsloAeroSec than OsloAero$_{def}$ most places in the PI atmosphere, while $N_{150}$ (e) and $N_{200}$ (g) are higher. This follows the mechanism explained in section 5.2, that lower NPF efficiency in

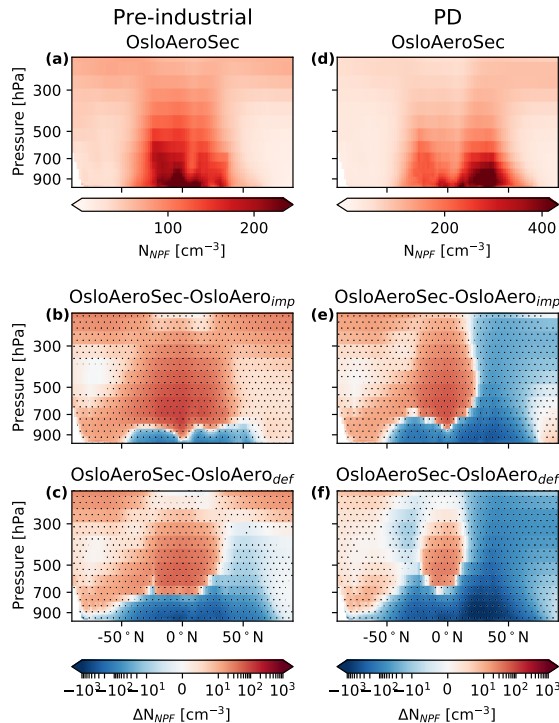

**Figure 8.** Top row: Zonally and annually averaged concentrations of $N_{NPF}$ for OsloAeroSec for PI (left) and PD (right). Rows 2–3: the absolute difference of OsloAeroSec to OsloAero$_{imp}$ (row 2) and OsloAero$_{def}$ (row 3) for PI (left) and PD (right), respectively. Dots are included in the plots to indicate where the difference between the two models is significant with a two-tailed paired Student's t–test with 95 % confidence interval.

OsloAeroSec leads to fewer, but larger particles. The higher concentrations in OsloAeroSec of e.g. $N_{200}$, comes from the modes shifting to higher median diameters when the number of NPF particles is lower. There is also a good correspondence between the difference in $N_{150}$ and/or $N_{200}$ and the difference in CDNC in most areas in the atmosphere. Note in for example the Amazon area, where much lower concentrations of $N_{100}$ (and NPF efficiency) are associated with much higher concentrations of $N_{200}$, but not $N_{150}$. That the CDNC is higher here, tells us that the activation diameter here is probably usually between 150–200 nm. Additionally, the supersaturation is higher in OsloAeroSec due to fewer particles that compete for the water vapor (see figure S20), which has a small positive impact on the number of particles which activate.

To investigate further these relationships between changes in $N_d$ and CDNC in the PI simulations, we compute the correlation between $\Delta$CDNC and $\Delta N_d$ where $\Delta$ signifies the difference between OsloAeroSec and OsloAero$_{def}$. First we compute the correlation between $\Delta$CDNC and $\Delta N_{NPF}$ over time and longitude, shown in Fig. 10c. This reveals that close to the surface,

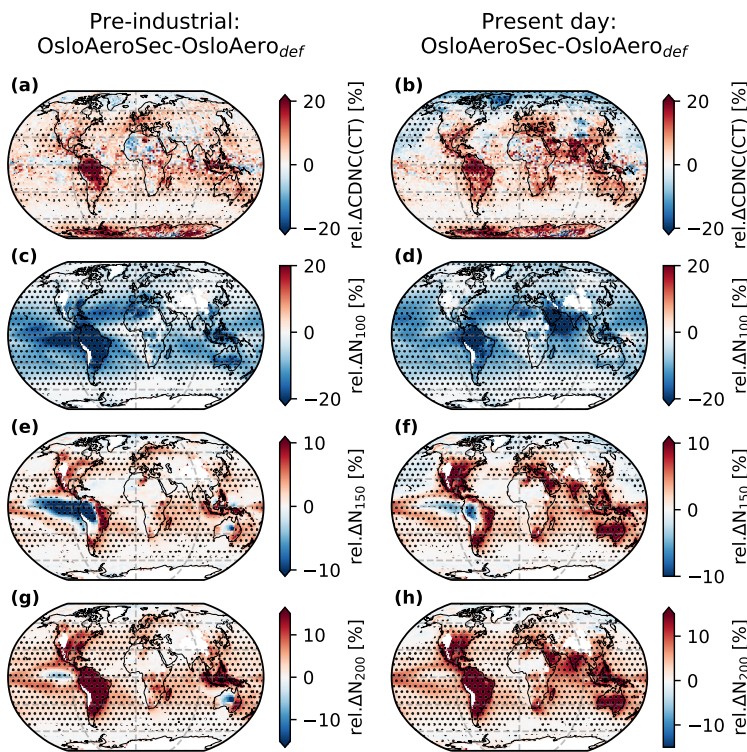

**Figure 9.** Top row: Relative difference in annual average cloud top cloud droplet number concentrations (CDNC(CT)) at cloud top between OsloAeroSec and OsloAero$_{def}$. Row 2–3: difference in average particle number concentration for particles larger than 100 nm (row 2), 150 nm (row 3) and 200 nm (row 4). The left column shows the difference for the pre-industrial atmosphere and the right column shows the difference for the present day atmosphere. The average particle concentrations are calculated by averaging up to 850 hPa and averaging by pressure difference. Dots are included in the plots to indicate where the difference between the two models is significant with a two-tailed paired Student's t–test with 95 % confidence interval.

$\Delta$CDNC and $\Delta N_{NPF}$ are mostly negatively correlated indicating that these areas, NPF inhibits activation. In remote regions, like e.g. the Southern Ocean or high in the free troposphere, there is a positive correlation between $\Delta N_{NPF}$ and $\Delta$CDNC, indicating that here we are in a NPF enhanced activation regime and relevant parts of the NPF mode particles activate.

Second, we compute the correlations between $\Delta$CDNC and $\Delta N_{50}$, $\Delta N_{100}$, $\Delta N_{150}$, $\Delta N_{200}$ and $\Delta N_{250}$ for different regions (see Table 5 for definitions) at different heights. These relationships for the PI simulations are shown in Fig. 11, column 1. If $\Delta$CDNC correlates clearly with the change in concentration of particles above some diameter $d$, $N_d$, this indicates that these

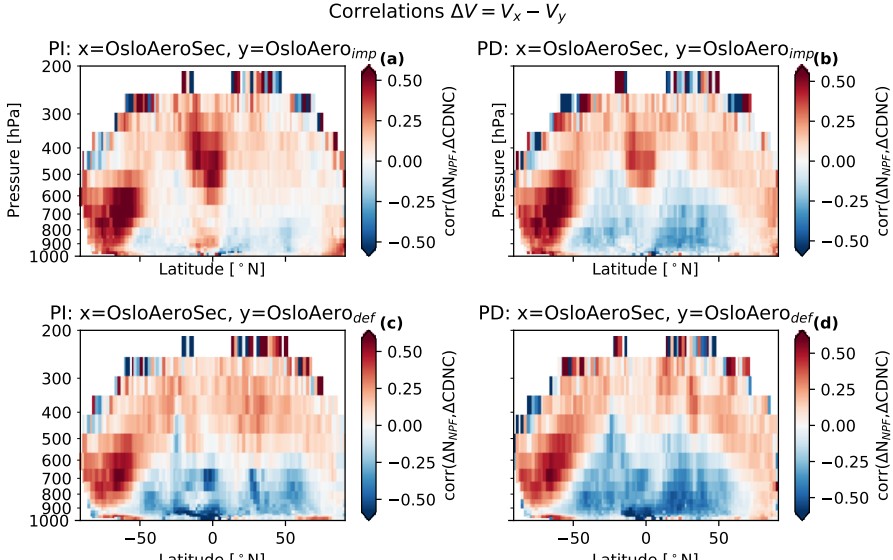

**Figure 10.** Correlations between the absolute difference in CDNC and the absolute difference in $N_{NPF}$ between the model versions, calculated from monthly mean files over time and longitude. The correlations from the difference between OsloAeroSec and OsloAero$_{imp}$ is shown in the top row. The correlations from the difference between OsloAeroSec and OsloAero$_{def}$ is shown in the bottom row. The correlations in the PI simulations are shown to the left and the ones for the PD simulations to the right.

**Table 5.** Region overview. These regions are used to create vertical average profiles.

| Region name | Latitudes | Longitudes |
|---|---|---|
| Global | All | All |
| Antarctic | 60–90 °S | 180 °W − 180 °E |
| Pacific S | 30 °N − 60 °N | 170°E − 120 ° W |
| Pacific N | 60 °S − 30 °S | 170°E − 140 ° W |

particle sizes are relevant for cloud droplet activation in the region. On the other hand if there is a negative correlation, this indicates that the particles are too small to activate.

Globally, Fig. 11a, show that CDNC correlates strongest with $N_{200}$ and $N_{250}$ close to the surface, with an anti-correlation with $N_{50}$ and $N_{100}$. The sign of the correlations switch at around $600\,\mathrm{hPa}$. In the relatively clean Antarctic (here defined as below 60 °S), the correlation is positive with the smaller particles, i.e. $N_{50}$, throughout the atmosphere. This indicates that NPF enhances activation in Antarctica and that the number of particles dominates, rather than the size of the particles. Figures 11e and g show the South and North Pacific, and are included because they show opposite sign in CDNC for the PD simulations and we will discuss them further in the next section. In the PI simulations, however, the South Pacific shows a clear correlation

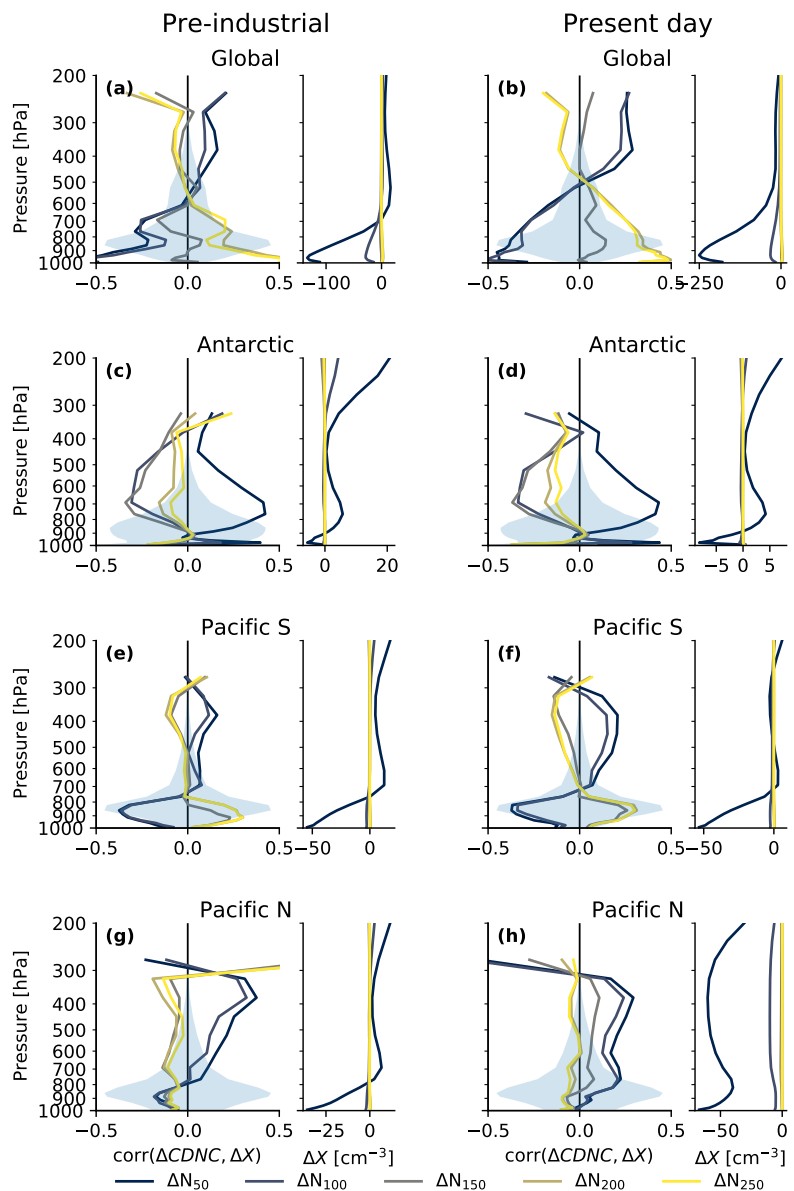

**Figure 11.** Left panel of each subplot: Correlations by pressure level between the absolute difference between OsloAero$_{def}$ and OsloAeroSec in cloud droplet number concentration ($\Delta$CDNC) and the absolute difference in number of particles with diameters above 50, 100, 150, 200 and 250 nm for different regions. The blue shaded signifies the fractional occurrence of liquid cloud and is included to give an idea of where the aerosols may actually have a noticeable impact on clouds. The right panel of each subplot shows the change in the aerosol concentration for the relevant region. See Table 5 for definitions of regions.

with the larger particles (diameters larger than 150, 200 and 250), while in the North Pacific, the correlation is closer to zero or insignificant.

### 5.4.2 Summary hypothesis 3: Higher activation in the pre-industrial atmosphere

We do indeed see higher aerosol activation and higher CDNC with OsloAeroSec in the PI simulations. This is due to a combination of two things: 1) In pristine areas, NPF particles are likely to activate and lead to higher CDNC – i.e. NPF enhances activation. In these areas OsloAeroSec in general produces more NPF particles than OsloAero$_{def}$ and thus CDNC increases. 2) In areas with higher aerosol number concentrations, NPF particles are unlikely to activate and NPF inhibits cloud droplet activation due to reducing the size of the larger particles. In these regions, OsloAeroSec in general produces less NPF particles than OsloAero$_{def}$ and thus CDNC increases.

## 5.5 The present day atmosphere: model to model differences

We now move to consider differences in the PD simulations between OsloAeroSec and OsloAero$_{def}$ and will discuss the hypothesis 4, "Lower activation in PD".

While with PI emissions, there are large regions, especially at higher altitudes where OsloAeroSec produced more NPF particles than the other model versions. With PD emissions, these areas shrink, as the atmosphere becomes less pristine overall. This is seen in Fig. 7d-f (near surface average), and Fig. 8d-f (zonal average). Furthermore, it is interesting to see the impact of emissions in the Northern Hemisphere versus the Southern Hemisphere in the PD simulations. In the Northern Hemisphere, OsloAeroSec produces fewer particles than the other model versions at most heights and latitudes, while the opposite is the case for the Southern Hemisphere. This is likely due to a combination of much higher emissions and more vertical mixing in the Northern than Southern Hemisphere. In other words, larger parts of the Northern Hemisphere pass into a pollution level regime where the sectional scheme produces fewer particles than the others.

### 5.5.1 Cloud properties

Figure 9b shows that the difference in CDNC(CT) between OsloAeroSec and OsloAero$_{def}$ in the PD simulations. The Southern Hemisphere resembles the difference in PI (Fig. 9a) with widespread increase in CDNC. In the middle– to high northern latitudes, on the other hand, CDNC is lower in OsloAeroSec than in OsloAero$_{def}$, opposite of in the PI simulations. In these last pristine northern regions, more NPF particles in OsloAero$_{def}$ seem indeed to lead to higher CDNC than in OsloAeroSec.

Let us again consider the model to model difference in size distribution. Figure 9d, f and h, shows $\Delta N_{100}$, $\Delta N_{150}$ and $\Delta N_{200}$. Here we see that the pristine northern hemisphere $\Delta$CDNC resembles most the change in $N_{100}$, while in the Southern hemisphere, $\Delta$CDNC resembles more that of the larger particles ($N_{150}$ and $N_{200}$). Note especially how the polluted regions in the PD simulations around India and China have higher concentrations of $N_{200}$ and $N_{150}$ in OsloAeroSec than OsloAero$_{def}$ and corresponding higher CDNC. In these polluted regions, NPF in general inhibits cloud droplet activation because the activation diameter is large (bottom panel in Fig. 5). This is because there are many large particles which activate early and act as a sink

for water vapor, thus reducing $S_{max}$ and increasing the activation diameter (see Fig. S20b). On the other hand, the decreases in CDNC in OsloAeroSec compared to OsloAero$_{def}$ in the PD northern high latitudes correspond better to the change in the smaller particles, $N_{100}$ and partially $N_{150}$. This indicates that in these regions NPF enhances cloud droplet activation due to a smaller activation diameter (top panel in Fig. 5). Note that this is different in the PI and PD simulations: in the PD simulations, the CDNC goes down with OsloAeroSec in the northern high latitudes, in the PI it goes up. The reason for this is that the activation diameter depends both on the maximum supersaturation *and* the hygroscopicity. The hygroscopicity of the particles almost doubles from the PI to the PD, due to increased sulphate emissions (see Fig. S26). The more hygroscopic particles in the PD simulations can then activate at smaller diameters (given the same $S_{max}$). The regions where CDNC is enhanced by NPF thus spreads in the pristine northern latitudes, favoring cloud droplet activation in OsloAero$_{def}$ over OsloAeroSec. Mark that the difference in hygroscopicity is large between the PI and PD simulations (again, see S26), but small ($\sim 5\%$) between the different model versions.

It is thus clear that hygroscopicity plays a role, but only in terms of making the effect of NPF particles different in the PI and in the PD simulations, where with PD emissions the NPF particles are more likely to activate. In other words, because hygroscopicity increases in PD, the areas where NPF enhances cloud activation expand in the PD northern hemisphere compared to the pre-industrial atmosphere.

Let us again consider the correlations between $\Delta$CDNC and $N_{NPF}$, $\Delta N_{50}$, $\Delta N_{100}$, $\Delta N_{150}$, $\Delta N_{200}$ and $\Delta N_{250}$ for different regions, shown for the PD atmosphere in Fig. 10 and Fig. 11b, d, f and h.

Globally, the correlation of $\Delta$CDNC with the change in larger particles is more pronounced in the PD than the PI simulations (Fig. 11b and Fig. 10d), possibly indicating a stronger super saturation adjustment (reduced $S_{max}$) with more polluted PD emission conditions, leading to a higher activation diameter.

Furthermore, we investigate the North and South Pacific separately in Figs. 11e–h, because these two show opposite sign in the PD simulations: in the North Pacific, OsloAeroSec has lower CDNC than OsloAero$_{def}$, while in the South Pacific OsloAeroSec has higher CDNC (see Fig. S9b). In the South Pacific (e and f), the CDNC correlates best with the larger particles (diameter above $150\,\mathrm{nm}$) in both PI and PD. In the North Pacific on the other hand, the correlation is not clear for any particle number in the PI (g) and slightly positive for the smaller particles sizes in PD (h). The likely cause for the difference between the two cases is that 1) the South Pacific has higher concentrations of larger sea salt particles than the North Pacific (not shown), which can limit the maximum supersaturation and thus lead to a higher activation diameter, and 2) as mentioned above, the sulphate emissions are much higher in the PD Northern hemisphere, leading to more hygroscopic particles, and a lower activation diameter. In the South Pacific, we are therefore at the bottom panel of the sketch in Fig. 5, while in the North Pacific, we are more on the top panel. Note again that the hygroscopicity between the model versions with the same emissions (either with PI *or* PD emissions) changes very little (Fig. S26), which is why we only discuss changes between the PI and PD.

### 5.5.2   Summary hypothesis 4: Lower activation in the present day atmosphere

The discussion above shows that regionally, lower cloud droplet activation and CDNC with OsloAeroSec in the PD simulations, does indeed play a role in reducing the ERF$_{aci}$ in the pristine high northern latitudes and the North Pacific. Here the CDNC

is lower with OsloAeroSec than OsloAero$_{def}$ and thus OsloAero$_{def}$ has a stronger negative cloud radiative effect in the PD simulations. On the other hand, cloud droplet activation and CDNC in more polluted regions is higher with OsloAeroSec than OsloAero$_{def}$ (see Fig. 9b) in the PD simulations. This does, however, not have as big an impact on radiation (see e.g. Fig. S16) firstly because these areas are mostly continental and the cloud radiative effect is larger over dark ocean surfaces (e.g. the North Pacific) and secondly because the CDNC is already high in these regions with OsloAero$_{def}$ and thus the clouds are less susceptible to the increase to OsloAeroSec (see introduction for description of this effect). Furthermore, we have found that hygroscopicity changes from PI to PD plays a role by reducing the activation diameter and making NPF particles more likely to activate in the PD simulations compared to the PI. This means that the areas where NPF enhances cloud droplet activation expands and thus there are larger areas where OsloAero$_{def}$ has higher CDNC than OsloAeroSec. Both these factors result in a lower CDNC in the high northern latitudes with OsloAeroSec, and a corresponding lower magnitude in NCRE$_{Ghan}$.

## 5.6 Comparison to OsloAero$_{imp}$

We have mostly focused on the comparison of OsloAeroSec to OsloAero$_{def}$ in the above section, but there are important points to take away from comparing OsloAeroSec to OsloAero$_{imp}$ as well. Note that OsloAero$_{imp}$ has the same updates to oxidants and nucleation rate as OsloAeroSec, but does not have the sectional scheme. Also, remember that OsloAero$_{imp}$ has much lower NPF efficiency than OsloAero$_{def}$, but compared to OsloAeroSec it is more similar, but depends on the region. In general OsloAeroSec produces more NPF particles in pristine regions, while OsloAero$_{imp}$ produces more particles in regions with higher aerosol concentrations.

When comparing only OsloAeroSec and OsloAero$_{def}$, it is not possible to separate the effect that increased NPF efficiency in remote regions has from decreased NPF efficiency in high-aerosol regions with respect to the ERF$_{aci}$. It is perhaps tempting to think that the reduction in NPF efficiency is alone responsible for the overall effect, and that the increase in NPF efficiency in remote regions is negligible. If so, any scheme which reduced NPF efficiency would have the same effect. The OsloAero$_{imp}$ simulation however, represents exactly such another scheme which reduces the NPF efficiency compared to OsloAero$_{def}$, with roughly the same amount as OsloAeroSec, though without the increases in NPF efficiency in remote regions. However, OsloAero$_{imp}$ does not weaken ERF$_{aci}$ like OsloAeroSec does, but rather slightly strengthens it. In essence, this shows that it is the combination of decreasing NPF efficiency in high aerosol regions and increasing NPF efficiency in low-aerosol regions which together gives the weakened ERF$_{aci}$ in OsloAeroSec.

## 5.7 Summary of hypothesis

We now summarize and relate the results back to the hypothesis presented in section 5.2.

1 **Smaller $\Delta_{\text{PD-PI}}N_a$:** While it is true that $N_a$ increases less from PI to PD with OsloAeroSec than OsloAero$_{def}$ (and OsloAero$_{imp}$), this can only explain the results in remote regions. Furthermore, OsloAero$_{imp}$ offers as a counter argument against this hypothesis: it also has a lower PD to PI change in $N_a$ ($\Delta_{\text{PD-PI}}N_a$) than OsloAero$_{def}$, but contrary to

OsloAeroSec, OsloAero$_{imp}$ has a stronger negative ERF$_{aci}$ than OsloAero$_{def}$. In sum, this hypothesis does not explain well the differences in ERF$_{aci}$.

2 **Higher N$_a$ in PI:** OsloAeroSec mostly produces fewer particles than OsloAero$_{def}$ in the PI simulations and this is thus only true in remote regions. This hypothesis can therefore not explain the resulting ERF$_{aci}$.

3 **Higher cloud droplet activation in PI:** We found that OsloAeroSec has higher CDNC than the other model versions in the PI simulations both due to more efficient NPF in remote regions where NPF enhances cloud droplet activation (small activation diameter) and due to less efficient NPF in regions where NPF inhibits cloud droplet activation (large activation diameter). In these last areas, OsloAeroSec indeed has a higher concentration of larger particles than OsloAero$_{def}$ and OsloAero$_{imp}$, due to the condensate being distributed to fewer particles in OsloAeroSec. This hypothesis therefore explains well the part of the change in ERF$_{aci}$ originating from difference in NCRE$_{Ghan}$ in the PI simulations.

4 **Lower cloud droplet activation in PD:** We found this hypothesis to play an important role in the northern high latitudes, especially the North Pacific, were sulphate emissions are high in the PD simulations. Due to higher hygroscopicity in the PD simulations compared to the PI, the NPF particles are more likely to activate (smaller activation diameter) and thus the number of particles (which is lower in OsloAeroSec) is more important than the particles sizes. This hypothesis therefore is important to explain the changes in the PD simulations.

Additionally, after the analysis of the results, we may add two more explanations:

5 **Hygroscopicity:** As explained for hypothesis 4 above, the change in hygroscopicity from PI to PD, results in larger areas in the northern pristine latitudes having a NPF enhanced cloud droplet activation regime in the PD simulations, compared to the PI. This results in stronger NCRE$_{Ghan}$ with OsloAero$_{def}$ than OsloAeroSec in the PD simulations which further leads to a stronger ERF$_{aci}$ in OsloAero$_{def}$ than OsloAeroSec.

6 **Regional differences:** The comparison with OsloAero$_{def}$ shows that regional differences in NPF matter significantly. For reasons discussed above, OsloAeroSec gives higher CDNC in the PI simulation in regions with susceptible clouds and large ERF$_{aci}$, which dominates the global average.

# 6 Implications and discussion

The results in this paper go in line with previous work which shows both that the ERF$_{aci}$ is sensitive to the PI aerosol characteristics, e.g. Carslaw et al. (2013), and that changes the NPF parameterization can highly influence ERF$_{aci}$ (e.g. Gordon et al., 2016). However, the reduction in ERF$_{aci}$ found with OsloAeroSec in our simulations, is not a result of increased NPF in under PI conditions alone. Rather the increase in CDNC and NCRE$_{Ghan}$ in the PI simulation originates from increased NPF efficiency where the NPF enhances cloud droplet activation, and decreased NPF efficiency where NPF inhibits particle

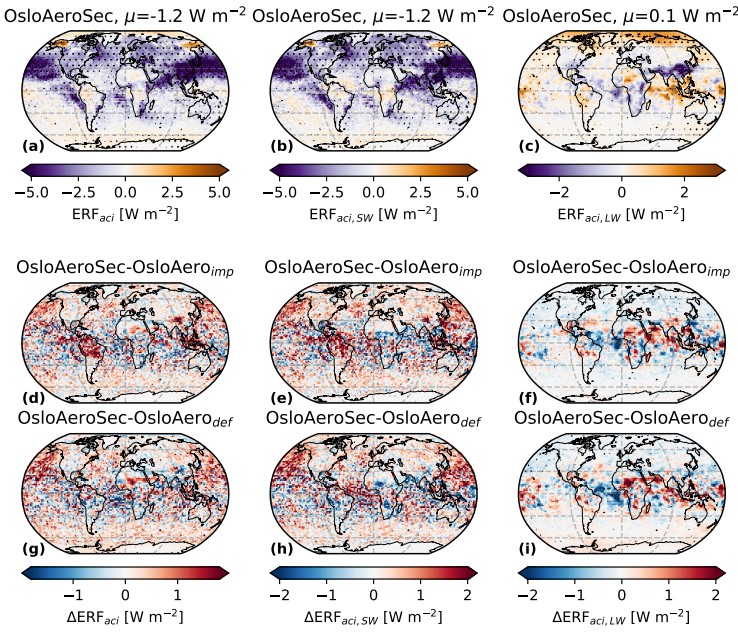

**Figure 12.** Annual averages of the $ERF_{aci}$ (left column), the short wave component of $ERF_{aci}$, $ERF_{aci,SW}$ (middle column) and the long wave component of the $ERF_{aci}$, $ERF_{aci,LW}$ (right column). The top panel shows the the absolute values for OsloAeroSec while the second and third row shows the difference OsloAero$_{imp}$ minus OsloAeroSec (second row) and OsloAero$_{def}$ minus OsloAeroSec (third row). Dots are included in the plots to indicate where the difference between the two models is significant with a two-tailed paired Student's t–test with 95 % confidence interval.

activation. Additionally, we find that the modelled increase in hygroscopicity from PI to PD from increased sulphate emissions, results in a lower activation diameter and thus that more of the NPF particles contribute to CDNC.

The effect of NPF inhibition on cloud droplet activation, was also found by Sullivan et al. (2018), where they modelled the NPF effect on clouds over the mid-western USA using WRF-Chem v3.6.1 and using a 20 bin sectional aerosol scheme (Model for Simulating Aerosol Interactions and Chemistry, MOSAIC). As in this study, they find that the growth of the larger particles are inhibited by the increased condensation sink from the NPF particles. That fact that the same effect is seen in simulations with a completely differently structured aerosol model, shows it to be unlikely that this is an artifact of the OsloAero model. However, their study uses the same activation scheme, Abdul-Razzak and Ghan (2000), and we cannot exclude that this scheme for example overestimates the supersaturation adjustment effect.

It is intrinsically difficult to directly evaluate the cloud-aerosol interactions in the models versus reality. This is partly because we cannot measure the pre-industrial atmosphere, but also due to the noisy nature of clouds. The evaluation of the model versions used in this study therefore focused on particles in sizes relevant for cloud activation and was primarily done in

Blichner et al. (2021). We have added further validation, using three datasets (Andreae et al., 2015; Wofsy et al., 2018; Andrade et al., 2015) representing different parts of the atmosphere than the previous comparison, to the supplementary of this study (see Supplementary, section 2). Overall, the sectional scheme shows significant improvement in the representation of particles in the CCN size range, and this indicates that our results for $\mathrm{ERF}_{aci}$ here represent an improvement.

As mentioned in the section 2, the sectional scheme, OsloAeroSec, has a higher contribution from organics to the growth from 5 nm than OsloAero$_{def}$ and OsloAero$_{imp}$ (only ELVOC in OsloAero). One could argue that this may be the driving factor of all these results, but in fact this is not the case. We did a test run where organics were treated in the same way in OsloAeroSec as in OsloAero$_{def}$ and OsloAero$_{imp}$ and the result in terms on particle number changes very little (see Fig. S27 and Fig. S28).

We also investigated the sensitivity of $\mathrm{ERF}_{aci}$ to changes in the nucleation rate with both the original model and with the sectional scheme. This investigation in detailed in the Supplementary, section 1. Overall the results show that the change in $\mathrm{ERF}_{aci}$ between the sectional and default model is very resistant to changes in nucleation rate. There are small differences within the OsloAero model versions and within the OsloAeroSec versions based on the nucleation rate, but larger differences between the two groups.

Note that we have not discussed CCN concentrations in this discussion. There are two reasons for this: Firstly, these are not yet available as standard output for CAM6-Nor. Secondly, the CCN concentrations at a given supersaturation matters only when this supersaturation is actually achieved, so focusing on CDNC gives a more complete picture which is closer related to the actual climatic impact of the particles in question.

These results also illustrate the importance of adequately representing activation when investigating the effect of NPF on climate, and not simply considering CCN at fixed supersaturation as this will omit not only regional changes in updraft velocities, but also supersaturation adjustment by the aerosol population.

## 7   Conclusions

In this study, we have shown that including a sectional scheme (OsloAeroSec) for the growth of particles from nucleation and up to the original modal scheme, reduces the estimated $\mathrm{ERF}_{aci}$ by between 0.13-0.14 $\mathrm{Wm}^{-2}$. The reduction originates from higher CDNC and $\mathrm{NCRE}_{Ghan}$ in the PI simulation, together with a smaller increase from PI to PD. By comparing model versions with different NPF parameterization in the pre-industrial and present day atmosphere respectively, we find that NPF in fact inhibits cloud droplet activation in parts of the atmosphere and leads to lower CDNC, due to reducing the growth of the larger, primary particles. The overall $\mathrm{ERF}_{aci}$ therefore, depends on in which regions NPF is high/low both in the PI and in the PD simulations. The reduction in $\mathrm{ERF}_{aci}$ with OsloAeroSec originates partly from higher NPF efficiency in PI areas where NPF enhances cloud droplet activation and lower NPF efficiency in PI areas where NPF inhibits cloud droplet activation. Furthermore, we find that the increase in sulphate from the PI to the PD simulation increases the hygroscopicity of the particles and thus allows more NPF particles to activate. This expands the areas where NPF enhances cloud droplet activation in the PD simulations which also contributes to a weaker $\mathrm{ERF}_{aci}$ for OsloAeroSec than OsloAero$_{def}$.

Roughly speaking, we can say that the results in ERF$_{aci}$ originate from OsloAeroSec is adding particles where the NPF particles are likely to act as CCN and removing them where they are unlikely to activate directly and rather act to diminish the size of the other particles.

Overall, this study shows that a more physical representation of the early growth of particles results in a lower ERF$_{aci}$ and that adequately representing early growth on a regional scale is important when estimates of ERF$_{aci}$.

*Code availability.* The model code of NorESM2, release 2.0.1, is available at https://doi.org/10.5281/zenodo.3760870 (Seland et al., 2020). The code modifications in OsloAeroSec are available at https://doi.org/10.5281/zenodo.4265057 (Blichner, 2020), see Blichner et al. (2021) for details. The post-processing code for creating the figures in this paper are available at https://doi.org/10.5281/zenodo.5559026 (Blichner, 2021b).

*Data availability.* The model output is available for download at https://doi.org/10.11582/2021.00087 Blichner (2021a). Measurement
data used in the supplementary is available for download as follows: The Chacataya data is available for download from the EBAS database (http://ebas.nilu.no/Pages/DataSetList.aspx?key=24F2E44259AE4E089EC74128B761A2F8, last accessed 2021-08-15). The data from ATTO tower was downloaded from the Laboratory of Atmospheric Physics (LFA, IF-USP)) http://ftp.lfa.if.usp.br/ftp/public/LFA_ Processed_Data/T0a_ATTO/Level3/SMPS_2014toNov2020_ATTO_60m_InstTower/, last accessed 2021-08-15). The ATom data was accessed from https://daac.ornl.gov/ATOM/guides/ATom_merge.html (last accessed: 2021-06-20) (WOFSY and Team, 2018; Wofsy et al.,
2018).

*Author contributions.* SMB did the model code development and performed the simulations with NorESM. SMB did the data analysis and wrote the manuscript. SMB, MKS and TKB contributed with discussions regarding the experimental design and data analysis. All contributors have contributed to the discussions regarding the manuscript.

*Competing interests.* There are no competing interests.

*Acknowledgements.* Many thanks to Dirk Oliviè and Alf Kirkevåg at Meteorologisk institutt for answering the many questions and thanks to Dirk Oliviè who let us use his simulations as initialization for simulations. Thanks to Diego Aliaga for helping to design the schematic in figure 5. For the model evaluation (see supplementary), we would like to thank the science teams providing the data: We thank the Amazon Tall Tower Observatory (ATTO) science team for providing the SMPS dataset from the site. We thank the Chacaltaya/GAW station science team for providing the SMPS dataset from the site. Finally we thank the ATom team for providing the dataset and especially Christina
Williamson, Charles Brock and Agnieszka Kupc for helpful comments on the analysis. This work was funded under the LATICE strategic

research initiative funded by the Faculty of Mathematics and Natural Sciences at the University of Oslo. This work has been financed by the Research Council of Norway (RCN) through the NOTUR/Norstore project NN2806K and NS9066K.

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
