# Peer review of "Reduced effective radiative forcing from cloud-aerosol interactions $(ERF_{aci})$ with improved treatment of early aerosol growth in an Earth System Model"

_Atmospheric Chemistry and Physics, 2021_

## Author Response (AR1)

**Authors response**

Sara M. Blichner[1], Moa K. Sporre[2] and Terje K. Berntsen[1]

[1]Departmenet of Geosciences and Centre for Biogeochemistry in the Anthropocene, University of Oslo, Oslo, Norway
[2]Department of Physics, Lund University, Lund, Sweden

Correspondence: Sara M. Blichner (s.m.blichner@geo.uio.no / sara.blichner@aces.su.se )

*We thank you for very helpful and insightful comments which has resulted in a, in our opinion, greatly improved manuscript.*

*We present our responses to the questions and comments below. The comments from the reviewer are written first, and our responses follow in italic. We will answer the comments from both referees in this document.*

**Referee 1:**

Blichner et al. have developed, in previous work, a more sophisticated treatment of small secondary aerosol particles that form via new particle formation (NPF) from the gas phase and numerically resolves the evolving size distribution for small particles in discrete size bins. Blichner et al. report from their previous study that the revised parameterisation is superior to the one typically used in the NorESM in comparison to observational data.

This present work is entirely a modelling study, no observational data are shown at all. It considers a range of uncoupled runs (ocean and sea ice prescribed) with nudging of atmospheric dynamics. For three different model versions (standard, revised parameterisation of small aerosols, and an intermediate version) present-day and pre-industrial simulations are conducted and compared in terms of radiative fluxes, drop number, and aerosol number.

The overall result is a small (~10%) reduction in the ERF due to aerosol-cloud interactions with the revised parameterisation. The bulk of the study is then dedicated to explaining how various processes lead to this net effect in the model. This analysis is very diligently performed and very well explained. It is plausible. It is a pity that nowhere observations are used to try and evaluate to which extent the modelled and hypothesized effects may reflect reality, but I acknowledge this is difficult to do since the effects are small, regionally very variable, and hard

to measure (in particular of course there are no measurements for the pre-industrial atmosphere).

Nevertheless, I believe the study is interesting enough for the readership of Atmos. Chem. Phys. I do not have many recommendations.

- *We thank the reviewer for these comments. It is indeed difficult to validate the effects, though we have evaluated the aerosol concentrations in these same model version in Blichner et al., (2021) and we have now added some further validation to the supplementary of this manuscript.*

l6 superfluous "with"? → "contributes a large"

- *Good suggestion. The manuscript has been changed accordingly.*

l10 radius or diameter?

- *Good point, we have added "5--39.6 nm **in diameter**".*

l15 sounds to me like this is the same number

- *The line in question reads: "We find that the ERFaci with the sectional scheme is −1.16 Wm−2, which is 0.13 Wm−2 weaker compared to the default scheme."*
- *Despite our best efforts, we cannot seem to see which numbers are the same?*

l38 "which is dependent"

- *Indeed, the manuscript has been changed accordingly.*

l48 this concerns of course only the secondary aerosol

- *Yes, good point, we have added "secondary" so it now reads: "will determine if the **secondary** aerosol mass is distributed as very few, very large particles or many smaller particles."*

l55 radius or diameter?

- *Good point, this should be diameter: it now reads "(⍰10 nm **in diameter**)"*

l58 "proportional"

- *Indeed. The manuscript has been changed accordingly.*

l59 "dependent on" rather "describing"?

- *Good suggestion, the manuscript has been changed accordingly.*

l64 It does not seem obvious that one cannot implicitly take into account time-varying conditions. Do the authors perhaps mean, that current parameterisations use such an assumption?

- *The sentence reads: "This is mainly because parameterizing the growth means assuming steady state conditions during the growth, i.e. that growth rate and coagulation sink are constant, there is no mixing and so on. This assumption is usually not appropriate, especially since the growth can take many hours or even grow over days."*
- *Yes, we mean to say that the parameterizations used in NorESM makes such an assumption. It is not really clear to us how such parameterizations could take into account time varying conditions, given that the changes in conditions may well be dependent on other factor than the aerosols themselves (changes emissions, chemistry, meteorology etc.) and taking changes into account without explicitly modelling, seems like more of a best-guess approach.  To make this clearer, we have changed the sentence to:*
- *"This is mainly because these parameterizations assume steady state conditions during the growth, i.e. that growth rate and coagulation sink are constant, and **changes in chemistry, mixing or emissions cannot be taken into account.** This assumption is usually not appropriate, especially since the growth can take many hours or even days."*

l73 This is for number concentrations presumably

- *The sentence as is reads "In other words, the if sulphuric acid emissions in the pre-industrial and present day atmosphere were the same, the pre-industrial atmosphere would have higher sulphuric acid concentrations because the condensation sink will be lower."*
- *We are not sure we understand what the reviewer means here, but we agree that the sentence could be clearer and have modified it to the following:*
- *"In other words, if an aerosol precursor species were to have the same emissions/production in the pre-industrial and present day atmosphere, the pre-industrial atmosphere would have higher gas phase concentrations because the condensation sink would be lower."*

l108 "in that its aerosol scheme"

- *Thanks! The manuscript has been changed accordingly.*

l111 "by Blichner et al"

- *Thanks! The manuscript has been changed accordingly.*

l144, l147 These numbers seem potentially rather important for the conclusions of the present manuscript. Where do they stem from? How sensitive are the results to this choice?

- *Good point! We have added the following sentences:*
- *"The yields used here are similar to those used in other global models (see e.g. Tsigaridis et al., 2014; Sporre et al., 2020; Dentener et al., 2006). All these yields are subject to substantial uncertainty (Shrivastava et al., 2017) – see e.g. Sporre et al. (2020) for an extensive discussion on the sensitivities to these choices."*

l505 "lower than"

- *Yes, indeed. Here we have also improved the sentence slightly from "it also has a ΔPD-PINa than OsloAerodef" to "it also has a lower PD to PI change in Na(ΔPD-PINa ) than OsloAerodef"*

**Referee 2:**

In this manuscript, the authors studied the effect of improved treatment of early aerosol growth (from 5 nm to 40 nm, with a sectional scheme) on effective radiative forcing associated with cloud-aerosol interactions (ERFaci) in the Norwegian Earth System Model v2 (NorESM2). Compared to the default scheme (OsloAerodef) that parameterizes the growth of freshly nucleated particles of a few nanometers and to the smallest mode in the model (>~40 nm), the explicit sectional treatment of this early growth (OsloAeroSec) enables the model to capture the variations in atmospheric condition and particle sizes during the growth that may take multiple hours. The results presented in this manuscript show that the ERFaci is sensitive to both the aerosol properties in the pre-industry (PI) simulation and parameterization of new particle formation (NPF), which are consistent with previous publications. The authors find that the ERFaci with the sectional scheme is 0.13Wm−2 weaker compared to the default scheme, resulting from OsloAeroSec producing more particles than the default scheme in pristine, low-aerosol concentration areas and less NPF particles in high-aerosol areas. The authors also show that NPF inhibits cloud droplet activation in high-aerosol-concentration regions but enhances cloud droplet activation in pristine/low aerosol regions, as a result of the difference in the cloud droplet activation sizes and the competition of condensing vapors between NPF particles and larger particles.  This manuscript deals with the treatment of aerosol formation and growth processes and their impacts on ERFaci in an Earth System Model,

which is important and is in the areas covered by ACP. The following comments should be addressed before I can recommend it for final publication in ACP.

1) This work focuses on the effect of aerosol growth treatment. A comparison of simulated aerosol properties based on both aerosol schemes with observations is essential but is lacking in the present manuscript. Does OsloAeroSec indeed improve the simulated aerosol sizes and number concentrations that are important for aerosol-cloud interaction? Are the spatial and temporal variations of simulated aerosol properties consistent with observations? A large amount aerosol measurements are currently available for model validations (for example, see Fanourgakis et al., https://doi.org/10.5194/acp-19-8591-2019, 2019).

- *Indeed! And we could have emphasized more that this was covered in our previous paper Blichner et al. (2021). We showed here that OsloAeroSec indeed improves the representation of aerosols of CCN relevant sizes (above 50 nm) in 24 stations.*
- *However, we have added a section in the supplementary with additional model evaluation against 3 extra datasets. These datasets were chosen in order to fill in the gaps in the previous validation and are thus from the tropical region (the ATTO tower and Mount Chacataya) and aircraft measurements (ATom). Both the ATom dataset and partially Chacataya also represent measurements of the free troposphere.*
- *We have also added the following paragraph to section 6 in the manuscript (the discussion):*
- *"It is intrinsically difficult to evaluate directly the cloud-aerosol interactions in the models versus reality. This is of course firstly because we cannot measure the pre-industrial atmosphere, but also due to the noisy nature of clouds, with many factors in play, and a lack of data. The model evaluation of the model versions used in this study therefore focused on particles in sizes relevant for cloud activation and was primarily done in Blichner et al. (2021). We have further added validation for three datasets representing different parts of the atmosphere than the previous one to the supplementary of this study (see Supplementary, section 2). Overall, the sectional scheme shows significant improvement in the representation of particles in the CCN size range, and this indicates that our results for ERFaci here represent an improvement"*

2) I think that the nucleation scheme used in this study (Lines 152-153) is oversimplified and likely does not give the correct spatial and temporal variations of NPF which is critical for the present study. Eq. 18 of Paasonen et al (i.e., J2=As1[H2SO4]+As2[org]) does not consider the well-recognized temperature dependence of nucleation rates (e.g., Yu et al., https://doi.org/10.5194/acp-17-4997-2017, 2017). This parameterization significantly overestimates the NPF rates in the summer and in the tropic regions (including Amazon). At least, the authors should take into account the temperature dependence and do a reasonable validation of the model simulations in terms of spatial and seasonal variations of particle number concentrations. The manuscript can be further improved if the sensitivity of the ERFaci based on two aerosol schemes to the nucleation parameterizations can be presented.

- *See the next comment for more information on the nucleation schemes! Both Paasonen et al., (2010) and Riccobono et al., (2014) are used in ESMs, but we do recognize the need to test the sensitivity of our results to the nucleation parameterization. We have therefore added a section to the supplementary describing the three new runs with changes to the nucleation scheme: two runs implementing the temperature dependency from Yu et al., (2017) both in the original model, OsloAero, and together with the sectional scheme, OsloAeroSec.  We have also added a third run using the Paasonen et al., (2010) parameterization instead of Riccobono et al., (2014) with the sectional scheme. The resulting ERF changes very little between the different versions together with the sectional scheme. Adding the temperature dependency in OsloAero$_{imp}$ leads to a somewhat stronger negative ERF, showing that the model without the sectional scheme may well be more sensitive to the nucleation rate.*

[Figure]

-
- *We have also added the following paragraph to the discussion:*
- *"We also investigated the sensitivity of ERFaci to changes in the nucleation rate with both the original model and with the sectional scheme. This investigation in detailed in the Supplementary, section 1. Overall the results show that the change in ERFaci between the sectional and default model is very resistant to changes in nucleation rate. There are small differences within the OsloAero model versions and within the OsloAeroSec versions based on the nucleation rate, but larger differences between the two groups."*

3) Figure 8a. Very high PI N_NPF in the tropic region is likely a result of the nucleation parameterization used (J2=As1[H2SO4]+As2[org]) and is against what is observed (for example over Amazon). As the authors acknowledge in the main text, PI aerosol characteristic is important for ERF$_{aci}$ calculation so it is essential to validate the PD aerosol simulations in pristine/low aerosol regions.

- *Due to an oversight on the authors part, the part of the model description concerning nucleation rate was missing. We apologize for this and have added it to the manuscript.*

- *The nucleation parameterization in OsloAeroSec (which is shown in figure 8a) is in fact Riccobono et al., (2014), not Paasonen et al., (2010).*
- *We have added the following sentence*
- *"The boundary layer nucleation parameterization has been updated from Paasonen et al. (2010) to Riccobono et al. (2014), and is now*
- *Jnuc = A3[H2SO4]2[ELVOC ] (3)*
- *where A3 = 3.27 × 10−21 cm6 s−1"*
- *The change is of course well explained in our previous model development paper (Blichner et al 2021), but it was clearly an error that it was not explained here too.*
- *Furthermore, we have added a table and text outlining the differences between the model versions*

**Table 1.** Model version overview.

| Simulation | Nucleation parameterization | Oxidant treatment | Early growth treatment |
|---|---|---|---|
| OsloAeroSec | $A_3[H_2SO_4]^2 \times [ELVOC]$ [*] | Improved diurnal variation | Lehtinen et al. (2007) + sectional scheme |
| OsloAero$_{imp}$ | $A_3[H_2SO_4]^2 \times [ELVOC]$ [*] | Improved diurnal variation | Lehtinen et al. (2007) |
| OsloAero$_{def}$ | $A_1[H_2SO_4] + A_2[ELVOC]$ [†] | Default diurnal variation | Lehtinen et al. (2007) |

$A_1 = 6.1 \times 10^{-7}$ s$^{-1}$
$A_2 = 3.9 \times 10^{-8}$ s$^{-1}$
$A_3 = 3.27 \times 10^{-21}$ cm$^6$s$^{-1}$
[*] Riccobono et al. (2014)
[†] Paasonen et al. (2010)

-
- *"In the result section we compare three different model versions, OsloAerodef , OsloAeroimp and OsloAeroSec. The first, OsloAeroSec, is the default model as used e.g. in the CMIP6 simulations, described in section 2.1.1 above. The third is with the sectional scheme, OsloAeroSec, as is decribed in section 2.1.2 and by Blichner et al. (2021). The second version, OsloAeroimp, is the default model but with the same changes to the nucleation scheme and the oxidant diurnal variation as are used in OsloAeroSec (described above). This is summarized in Table 1. The motivation for including all these model versions is to be able to distinguish the effect of the sectional scheme from that of the changes in nucleation and oxidants."*
- *We thank the reviewer for pointing this out and giving us a chance to explain.*

4) Lines 10, 162-163, and Fig 1 caption: The descriptions of the size range considered in the OsloAeroSec appear to be inconsistent. Please clarify. OsloAeroSec starts at 5 nm, right? Why not 2 nm?

- *We believe the reviewer is here referring to the fact that the smallest mode is 23.6 nm while the sectional scheme ends at 39.6 nm in diameter, where the particles are transferred to the modal scheme. This is because the number median diameter of the smallest mode is 23.6 nm, while the volume median diameter is 39.6 nm. Thus, in order to*

*preserve both mass and number, the particles are grown to 39.6 nm before added to the*
*sectional scheme. This is explained in the caption of Figure 1.*

- *To make this clearer, we have added the following sentences to the model description in section 2.1.2 (previous section 2.2)*
- *"The sectional scheme starts at 5 nm and extends to 39.6 nm, where the particles are transferred to the NPF mode in the pre-existing aerosol scheme. The sectional scheme extends to the volume median diameter (39.6 nm) rather than the number*
- *175 median diameter (23.6 nm) in order to preserve both number and mass during the transfer between the schemes."*
- *Secondly, the choice of adding the sectional scheme at 5 nm versus e.g. 2 nm is a balance between accuracy and computational cost. There may be reasons to revise this choice in the future, but in this paper we wanted to evaluate the model version already published in Blichner et al., (2021).*

**References:**

Blichner, S. M. *et al.* (2021) 'Implementing a sectional scheme for early aerosol growth from new particle formation in the Norwegian Earth System Model v2: comparison to observations and climate impacts', *Geoscientific Model Development*, 14(6), pp. 3335–3359. doi: 10.5194/gmd-14-3335-2021.

Paasonen, P. *et al.* (2010) 'On the roles of sulphuric acid and low-volatility organic vapours in  the initial steps of atmospheric new particle formation', *Atmos. Chem. Phys.*, 10(22), pp. 11223–11242. doi: 10.5194/acp-10-11223-2010.

Riccobono, F. *et al.* (2014) 'Oxidation Products of Biogenic Emissions Contribute to Nucleation of Atmospheric Particles', *Science*, 344(6185), pp. 717–721. doi: 10.1126/science.1243527.

Yu, F. *et al.* (2017) 'Impact of temperature dependence on the possible contribution of organics to new particle formation in the atmosphere', *Atmospheric Chemistry and Physics*, 17(8), pp. 4997–5005. doi: 10.5194/acp-17-4997-2017.